# How Does a Neural Network's Architecture Impact Its Robustness to Noisy Labels?

**Jingling Li**
Department of Computer Science
University of Maryland
College Park, MD 20740
`jingling@cs.umd.edu`

**Mozhi Zhang**
Department of Computer Science
University of Maryland
College Park, MD 20740
`mozhi@cs.umd.edu`

**Keyulu Xu**
Computer Science and Artificial Intelligence Laboratory (CSAIL)
Massachusetts Institute of Technology (MIT)
Cambridge, MA 02139
`keyulu@mit.edu`

**John Dickerson**
Department of Computer Science
University of Maryland
College Park, MD 20740
`john@cs.umd.edu`

**Jimmy Ba**
Department of Computer Science
University of Toronto, Canada
`jba@cs.toronto.edu`

## Abstract

Noisy labels are inevitable in large real-world datasets. In this work, we explore an area understudied by previous works — how the network's architecture impacts its robustness to noisy labels. We provide a formal framework connecting the robustness of a network to the alignments between its architecture and target/noise functions. Our framework measures a network's robustness via the predictive power in its representations — the test performance of a linear model trained on the learned representations using a small set of clean labels. We hypothesize that a network is more robust to noisy labels if its architecture is more aligned with the target function than the noise. To support our hypothesis, we provide both theoretical and empirical evidence across various neural network architectures and different domains. We also find that when the network is well-aligned with the target function, its predictive power in representations could improve upon state-of-the-art (SOTA) noisy-label-training methods in terms of test accuracy and even outperform sophisticated methods that use clean labels.

## 1 Introduction

Supervised learning starts with collecting labeled data. Yet, high-quality labels are often expensive. To reduce annotation cost, we collect labels from non-experts [1–4] or online queries [5–7], which are inevitably noisy. To learn from these noisy labels, previous works propose many techniques, including modeling the label noise [8–10], designing robust losses [11–14], adjusting loss before gradient updates [15–23], selecting trust-worthy samples [12, 22, 24–31], designing robust architectures [32–39], applying robust regularization in training [40–45], using meta-learning to avoid over-fitting [46, 47], and applying semi-supervised learning [28, 48–51] to learn better representations.

35th Conference on Neural Information Processing Systems (NeurIPS 2021).

While these methods improve some networks' robustness to noisy labels, we observe that their effectiveness depends on how well the network's architecture aligns with the target/noise functions, and they are less effective when encountering more realistic label noise that is class-dependent or instance-dependent. This motivates us to investigate an understudied topic: how the network's architecture impacts its robustness to noisy labels.

We formally answer this question by analyzing how a network's architecture aligns with the target function and the noise. To start, we measure the robustness of a network via the predictive power in its learned representations (Definition 1), as models with large test errors may still learn useful predictive hidden representations [52, 53]. Intuitively, the predictive power measures how well the representations can predict the target function. In practice, we measure it by training a linear model on top of the learned representations using a small set of clean labels and evaluate the linear model's test performance [54].

We find that a network having a more aligned architecture with the target function is more robust to noisy labels due to its more predictive representations, whereas a network having an architecture more aligned with the noise function is less robust. Intuitively, a *good* alignment between a network's architecture and a function exists if the architecture can be decomposed into several modules such that each module can simulate one part of the function with a *small* sample complexity. The formal definition of alignment is in Section 2.3, adapted from [55].

Our proposed framework provides initial theoretical support for our findings on a simplified noisy setting (Theorem 2). Empirically, we validate our findings on synthetic graph algorithmic tasks by designing several variants of Graph Neural Networks (GNNs), whose theoretical properties and alignment with algorithmic functions have been well-studied [55–57]. Many noisy label training methods are applied to image classification datasets, so we also validate our findings on image domains using different architectures.

Most of our analysis and experiments use standard neural network training. Interestingly, we find similar results when using DivideMix [49], a SOTA method for learning with noisy labels: for networks less aligned with the target function, the SOTA method barely helps and sometimes even hurts test accuracy; whereas for more aligned networks, it helps greatly.

For well-aligned networks, the predictive power of their learned representation could further improve the test performance of SOTA methods, especially on class-dependent or instance-dependent label noise where current methods on noisy label training are less effective. Moreover, on Clothing1M [58], a large-scale dataset with real-world label noise, the predictive power of a well-aligned network's learned representations could even outperform some sophisticated methods that use clean labels.

In summary, we investigate how an architecture's alignments with different (target and noise) functions affect the network's robustness to noisy labels, in which we discover that despite having large test errors, networks well-aligned with the target function can still be robust to noisy labels when evaluating their predictive power in learned representations. To formalize our finding, we provide a theoretical framework to illustrate the above connections. At the same time, we conduct empirical experiments on various datasets with various network architectures to validate this finding. Besides, this finding further leads to improvements over SOTA noisy-label-training methods on various datasets and under various kinds of noisy labels (Tables 5-10 in Appendix A).

## 1.1 Related Work

A commonly studied type of noisy label is the random label noise, where the noisy labels are drawn i.i.d. from a uniform distribution. While neural networks trained with random labels easily overfit [59], it has been observed that networks learn simple patterns first [52], converge faster on downstream tasks [53], and benefit from memorizing atypical training samples [60].

Accordingly, many recent works on noisy label training are based on the assumption that when trained with noisy labels, neural networks would first fit to clean labels [12, 25, 26, 49, 50] and learn useful feature patterns [18, 61–63]. Yet, these methods are often more effective on random label noise than on more realistic label noise (i.e., class-dependent and instance-dependent label noise).

Many works on representation learning have investigated the features preferred by a network during training [52, 64–66], and how to interpret or control the learned representations on clean data [54, 64, 67–69]. Our paper focuses more on the predictive power rather than the explanatory power

in the learned representations. We adapt the method in [54] to measure the predictive power in representations, and we study learning from noisy labels rather than from a clean distribution.

On noiseless settings, prior works show that neural networks have the inductive bias to learn simple patterns [52, 64–66]. Our work formalizes what is considered as a simple pattern for a given network via architectural alignments, and we extend the definition of alignment in [55] to noisy settings.

## 2 Theoretical Framework

In this section, we introduce our problem settings, give formal definitions for "predictive power" and "alignment," and present our main hypothesis as well as our main theorem.

### 2.1 Problem Settings

Let $\mathcal{X}$ denote the input domain, which can be vectors, images, or graphs. The task is to learn an underlying target function $f : \mathcal{X} \to \mathcal{Y}$ on a noisy training dataset $S := \{(\boldsymbol{x}_i, y_i)\}_{i \in \mathcal{I}} \bigcup \{(\boldsymbol{x}_i, \hat{y}_i)\}_{i \in \mathcal{I}'}$, where $y := f(\boldsymbol{x})$ denotes the true label for an input $\boldsymbol{x}$, and $\hat{y}$ denotes the noisy label. Here, the set $\mathcal{I}$ contains indices with clean labels, and $\mathcal{I}'$ contains indices with noisy labels. We denote $\frac{|\mathcal{I}'|}{|S|}$ as the *noise ratio* in the dataset $S$. We consider both regression and classification problems.

**Regression settings.** We consider a label space $\mathcal{Y} \subseteq \mathbb{R}$ and two types of label noise: a) **additive label noise** [70]: $\hat{y} := y + \epsilon$, where $\epsilon$ is a random variable independent from $\boldsymbol{x}$; b) **instance-dependent label noise**: $\hat{y} := g(\boldsymbol{x})$ where $g : \mathcal{X} \to \mathcal{Y}$ is a noise function dependent on the input.

**Classification settings.** We consider a discrete label space with $C$ classes: $\mathcal{Y} = \{1, 2, \cdots, C\}$, and three types of label noise: a) **uniform label noise**: $\hat{y} \sim \text{Unif}(1, C)$, where the noisy label is drawn from a discrete uniform distribution with values between $1$ and $C$, and thus is independent of the true label; b) **flipped label noise**: $\hat{y}$ is generated based on the value of the true label $y$ and does not consider other input structures; c) **instance-dependent label noise**: $\hat{y} := g(\boldsymbol{x})$ where $g : \mathcal{X} \to \mathcal{Y}$ is a function dependent on the input $\boldsymbol{x}$'s internal structures. Previous works on noisy label learning commonly study uniform and flipped label noise. A few recent works [71, 72] explore the instance-dependent label noise as it is more realistic.

### 2.2 Predictive Power in Representations

A network's robustness is often measured by its test performance after trained with noisy labels. Yet, since models with large test errors may still learn useful representations, we measure the robustness of a network by how good the learned representations are at predicting the target function — the predictive power in representations. To formalize this definition, we decompose a neural network $\mathcal{N}$ into different modules $\mathcal{N}_1, \mathcal{N}_2, \cdots$, where each module can be a single layer (e.g., a convolutional layer) or a block of layers (e.g., a residual block).

**Definition 1.** *(Predictive power).* Let $f : \mathcal{X} \to \mathcal{Y}$ denote the underlying target function where the input $\boldsymbol{x} \in \mathcal{X}$ is drawn from a distribution $\mathcal{D}$. Let $\mathcal{C} := \{(\boldsymbol{x}_i, y_i)\}_{i=1}^m$ denote a small set of clean data (i.e., $y_i = f(\boldsymbol{x}_i)$). Given a network $\mathcal{N}$ with $n$ modules $\mathcal{N}_j$, let $h^{(j)}(\boldsymbol{x})$ denote the representation from module $\mathcal{N}_j$ on the input $\boldsymbol{x}$ (i.e., the output of $\mathcal{N}_j$). Let $L$ denote the linear model trained with the clean set $\mathcal{C}$ where we use $h^{(j)}(\boldsymbol{x})$ as the input, and $y_i$ as the target value during training. Then the predictive power of representations from the module $\mathcal{N}_i$ is defined as

$$P_j(f, \mathcal{N}, \mathcal{C}) = \mathop{\mathbb{E}}_{\boldsymbol{x} \sim \mathcal{D}} \left[ l \left( f(\boldsymbol{x}), L(h^{(j)}(\boldsymbol{x})) \right) \right], \tag{1}$$

where $l$ is a loss function used to evaluate the test performance on the learning task.

**Remark.** Notice that smaller $P_j(f, \mathcal{N}, \mathcal{C})$ indicates better predictive power; i.e., the representations are better at predicting the target function. We empirically evaluate the predictive power using linear regression to obtain a trained linear model $L$, which avoids the issue of local minima as we are solving a convex problem; then we evaluate $L$ on the test set.

## 2.3 Formalization of Alignment

Our analysis stems from the intuition that a network would be more robust to noisy labels if it could learn the target function more easily than the noise function. Thus, we use architectural alignment to formalize what is easy to learn by a given network. Xu et al. [55] define the alignment between a network and a deterministic function via a sample complexity measure (i.e., the number of samples needed to ensure low test error with high probability) in a PAC learning framework (Definition 3.3 in Xu et al. [55]). Intuitively, a network aligns well with a function if each network module can easily learn one part of the function with a small sample complexity.

**Definition 2.** *(Alignment, simplified based on Xu et al. [55]).* Let $\mathcal{N}$ denote a neural network with $n$ modules $\mathcal{N}_j$. Given a function $f : \mathcal{X} \to \mathcal{Y}$ which can be decomposed into $n$ functions $f_j$ (e.g., $f(\boldsymbol{x}) = f_1(f_2(...f_n(\boldsymbol{x}))))$, the alignment between the network $\mathcal{N}$ and $f$ is defined via

$$Alignment(\mathcal{N}, f, \epsilon, \delta) := \max_j \mathcal{M}_{A_j}(f_j, \mathcal{N}_j, \epsilon, \delta), \tag{2}$$

where $\mathcal{M}_{A_j}(f_j, \mathcal{N}_j, \epsilon, \delta)$ denotes the sample complexity measure for $\mathcal{N}_j$ to learn $f_j$ with $\epsilon$ precision at a failure probability $\delta$ under a learning algorithm $A_j$.

**Remark.** Notice that smaller $Alignment(\mathcal{N}, f, \epsilon, \delta)$ indicates better alignment between network $\mathcal{N}$ and function $f$. If $f$ is obtuse or does not have a structural decomposition, we can choose $n = 1$, and the definition of alignment degenerates into the sample complexity measure for $\mathcal{N}$ to learn $f$. Although it is sometimes non-trivial to compute the exact alignment for a task without clear algorithmic structures, we could break this complicated task into sub-tasks, and it would be easier to measure the sample complexity of learning each sub-task.

Xu et al. [55] further prove that better alignment implies better sample complexity and vice versa.

**Theorem 1.** *(Informal; [55]) Fix $\epsilon$ and $\delta$. Given a target function $f : \mathcal{X} \to \mathcal{Y}$ and a network $\mathcal{N}$, suppose $\{x_i\}_{i=1}^M$ are i.i.d. samples drawn from a distribution $\mathcal{D}$, and let $y_i := f(x_i)$. Then $Alignment(\mathcal{N}, f, \epsilon, \delta) \leq M$ if and only if there exists a learning algorithm $A$ such that*

$$\mathbb{P}_{x \sim \mathcal{D}}\left[\|f_{\mathcal{N},A}(x) - f(x)\| \leq \epsilon\right] \geq 1 - \delta, \tag{3}$$

*where $f_{\mathcal{N},A}$ is the function generated by $A$ on the training data $\{x_i, y_i\}_{i=1}^M$.*

**Remark.** Intuitively, a function $f$ (with a decomposition $\{f_j\}_j$) can be efficiently learned by a network $\mathcal{N}$ (with modules $\{\mathcal{N}_j\}_j$) iff each $f_j$ can be efficiently learned by $\mathcal{N}_j$.

We further extend Definition 2 to work with a random process $\mathcal{F}$ (i.e., a set of all possible sample functions that describes the noisy label distribution).

**Definition 3.** *(Alignment, extension to various noise functions).* Given a neural network $\mathcal{N}$ and a random process $\mathcal{F}$, for each $f \in \mathcal{F}$, the alignment between $\mathcal{N}$ and $f$ is measured via $\max_j \mathcal{M}_{A_j}(f_j, \mathcal{N}_j, \epsilon, \delta)$ based on Definition 2. Then the alignment between $\mathcal{N}$ and $\mathcal{F}$ is defined as

$$Alignment^*(\mathcal{N}, \mathcal{F}, \epsilon, \delta) := \sup_{f \in \mathcal{F}} \max_j \mathcal{M}_{A_j}(f_j, \mathcal{N}_j, \epsilon, \delta),$$

where $\mathcal{N}$ can be decomposed differently for various $f$.

## 2.4 Better Alignment Implies Better Robustness (Better Predictive Power)

Building on the definitions of *predictive power* and *alignment*, we hypothesize that a network better-aligned with the target function (smaller $Alignment(\mathcal{N}, f, \epsilon, \delta)$) would learn more predictive representations (smaller $P_j(f, \mathcal{N}, \mathcal{C})$) when trained on a given noisy dataset.

**Hypothesis 1.** (Main Hypothesis). Let $f : \mathcal{X} \to \mathcal{Y}$ denote the target function. Fix $\epsilon$, $\delta$, a learning algorithm $A$, a noise ratio, and a noise function $g : \mathcal{X} \to \mathcal{Y}$ (which may be a drawn from a random process). Let S denote a noisy training dataset and $\mathcal{C}$ denote a small set of clean data. Then for a network $\mathcal{N}$ trained on $S$ with the learning algorithm $A$,

$$Alignment(\mathcal{N}, f, \epsilon, \delta) \downarrow \implies P_j(f, \mathcal{N}, \mathcal{C}) \downarrow, \tag{4}$$

where $j$ is selected based on the network's architectural alignment with the target function (for simplicity, we consider $j = n - 1$ in this work).

We prove this hypothesis for a simplified case where the target function shares some common structures with the noise function (e.g., class-dependent label noise). We refer the readers to Appendix C for a full statement of our main theorem with detailed assumptions.

**Theorem 2.** *(Main Theorem; informal) For a target function $f : \mathcal{X} \to \mathcal{Y}$ and a noise function $g : \mathcal{X} \to \mathcal{Y}$, consider a neural network $\mathcal{N}$ well-aligned with $f$ such that $P_j(f, \mathcal{N}, \mathcal{C})$ is small when training $\mathcal{N}$ on clean data (i.e., $P_j(f, \mathcal{N}, \mathcal{C}) < c$ for some small constant c). If there exists a function $h$ on the input domain $\mathcal{X}$ such that $f$ and $g$ can be decomposed as follows: $\forall x \in \mathcal{X}, f(\boldsymbol{x}) = f_r(h(\boldsymbol{x}))$ with $f_r$ being a linear function, and $g(\boldsymbol{x}) = g_r(h(\boldsymbol{x}))$ for some function $g_r$, then the representations learned by $\mathcal{N}$ on the noisy dataset still have a good predictive power with $P_j(f, \mathcal{N}, \mathcal{C}) < c$.*

We further provide empirical support for our hypothesis via systematic experiments on various architectures, target and noise functions across both regression and classification settings.

# 3 Experiments on Graph Neural Networks

We first validate our hypothesis on synthetic graph algorithmic tasks by designing GNNs with different levels of alignments to the underlying target/noise functions. We consider regression tasks. The theoretical properties of GNNs and their alignment with algorithmic regression tasks are well-studied [55–57, 73]. To start, we conduct experiments on different types of additive label noise and extend our experiments to instance-dependent label noise, which is closer to real-life noisy labels.

**Common Experimental Settings.** The training and validation sets always have the same noise ratio, the percentage of data with noisy labels. We choose mean squared error (MSE) and Mean Absolute Error (MAE) as our loss functions. Due to space limit, the results using MAE are in Appendix A.3. All training details are in Appendix B.3. The test error is measured by mean absolute percentage error (MAPE), a relative error metric.

## 3.1 Background: Graph Neural Networks

GNNs are structured networks operating on graphs with MLP modules [74–80]. The input is a graph $\mathcal{G} = (V, E)$ where each node $u \in V$ has a feature vector $\boldsymbol{x}_u$, and we use $\mathcal{N}(u)$ to denote the set of neighbors of $u$. GNNs iteratively compute the node representations via message passing: (1) the node representation $\boldsymbol{h}_u$ is initialized as the node feature: $\boldsymbol{h}_u^{(0)} = \boldsymbol{x}_u$; (2) in iteration $k = 1..K$, the node representations $\boldsymbol{h}_u^{(k)}$ are updated by aggregating the neighboring nodes' representations with MLP modules [81]. We can optionally compute a graph representation $\boldsymbol{h}_\mathcal{G}$ by aggregating the final node representations with another MLP module. Formally,

$$\boldsymbol{h}_u^{(k)} := \sum_{v \in \mathcal{N}(u)} \text{MLP}^{(k)}\Big(\boldsymbol{h}_u^{(k-1)}, \boldsymbol{h}_v^{(k-1)}\Big), \tag{5}$$

$$\boldsymbol{h}_\mathcal{G} := \text{MLP}^{(K+1)}\Big(\sum_{u \in \mathcal{G}} \boldsymbol{h}_u^{(K)}\Big). \tag{6}$$

Depending on the task, the output is either the graph representation $\boldsymbol{h}_\mathcal{G}$ or the final node representations $\boldsymbol{h}_u^{(K)}$. We refer to the neighbor aggregation step for $\boldsymbol{h}_u^{(k)}$ as *aggregation* and the pooling step for $\boldsymbol{h}_\mathcal{G}$ as *readout*. Different tasks require different aggregation and readout functions.

## 3.2 Additive Label Noise

Hu et al. [70] prove that MLPs are robust to additive label noises with zero mean, if the labels are drawn i.i.d. from a Sub-Gaussian distribution. Wu and Xu [82] also show that linear models are robust to zero-mean additive label noise even in the absence of explicit regularization. In this section, we show that a GNN *well-aligned* to the target function not only achieves low test errors on additive label noise with zero-mean, but also learns *predictive* representations on noisy labels that are drawn from non-zero-mean distributions despite having large test error.

**Task and Architecture.** The task is to compute the maximum node degree:

$$f(\mathcal{G}) := \max_{u \in \mathcal{G}} \sum_{v \in \mathcal{N}(u)} 1. \tag{7}$$

$$h_{\mathcal{G}} := \mathrm{MLP}^{(2)}\underbrace{(\max_{u \in \mathcal{G}}}_{\text{Module}^{(2)}} \underbrace{(\sum_{v \in \mathcal{N}(u)} \mathrm{MLP}^{(1)}(h_u, h_v))}_{\text{Module}^{(1)}}$$

$$f(\mathcal{G}) := \underbrace{max}_{\substack{u \in \mathcal{G} \\ f_2(\cdot)}} \underbrace{\sum_{\substack{v \in \mathcal{N}(u) \\ f_1(\cdot)}} 1}$$

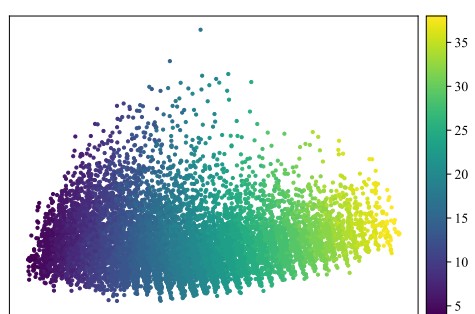

Figure 1: **Max-sum GNN aligns well with the task maximum degree.** Max-sum GNN $h_{\mathcal{G}}$ can be decomposed into two modules: Module$^{(1)}$ and Module$^{(2)}$, and the target function $f(\mathcal{G})$ can be similarly divided as $f(\mathcal{G}) = f_2(f_1(\mathcal{G}))$. As the nonlinearities of the target function have been encoded in the GNN's architecture, $f(\mathcal{G})$ can be easily learned by $h_{\mathcal{G}}$: $f_1(\cdot)$ can be easily learned by Module$^{(1)}$, and $f_2(\cdot)$ is the same as Module$^{(2)}$.

Figure 2: **PCA visualization of hidden representations** from a max-sum GNN trained with additive label noise drawn from $\mathcal{N}(10, 15)$ at 100% noise ratio. Each dot denotes a single training example and is colored with its true label. The x-axis and y-axis denote the projected values at the first and second principal components. As the colors change gradually from left to right, the largest principal component of the representations have a clear linear relationship with the true labels.

We choose this task as we know which GNN architecture aligns well with this target function—a 2-layer GNN with max-aggregation and sum-readout (max-sum GNN):

$$\boldsymbol{h}_{\mathcal{G}} := \mathrm{MLP}^{(2)}\Big(\max_{u \in \mathcal{G}} \sum_{v \in \mathcal{N}(u)} \mathrm{MLP}^{(1)}\Big(\boldsymbol{h}_u, \boldsymbol{h}_v\Big)\Big), \tag{8}$$

$$\boldsymbol{h}_u := \sum_{v \in \mathcal{N}(u)} \mathrm{MLP}^{(0)}\Big(\boldsymbol{x}_u, \boldsymbol{x}_v\Big). \tag{9}$$

Figure 1 demonstrates how exactly the max-sum GNN aligns with $f(\mathcal{G})$. Intuitively, they are well-aligned as the MLP modules of max-sum GNN only need to learn simple constant functions to simulate $f(\mathcal{G})$. Based on Figure 1, we take the output of Module$^{(2)}$ as the learned representations for max-sum GNNs when evaluating the predictive power.

**Label Noise.** We corrupt labels by adding independent noise $\epsilon$ drawn from three distributions: Gaussian distributions with zero mean $\mathcal{N}(0, 40)$ and non-zero mean $\mathcal{N}(10, 15)$, and a long-tailed Gamma distribution with zero-mean $\Gamma(2, 1/15) - 30$. We also consider more distributions with non-zero mean in Appendix A.2.

**Findings.** In Figure 3, while the max-sum GNN is robust to *zero-mean* additive label noise (dotted yellow and purple lines), its test error is much higher under non-zero-mean noise $\mathcal{N}(10, 15)$ (dotted red line) as the learned signal may be "shifted" by the non-centered label noise. Yet, max-sum GNNs' learned representations under these three types of label noise all predict the target function well when evaluating their predictive powers with 10% clean labels (solid lines in Figure 3).

Moreover, when we plot the representations (using PCA) from a max-sum GNN trained under 100% noise ratio with $\epsilon \sim \mathcal{N}(10, 15)$, the representations indeed correlate well with true labels (Figure 2). This explains why the representation learned under noisy labels can recover surprisingly good test performance despite that the original model has large test errors.

The predictive power of randomly-initialized max-sum GNNs is in Table 3 (Appendix A.1).

### 3.3 Instance-Dependent Label Noise

Realistic label noise is often instance-dependent. For example, an option is often incorrectly priced in the market, but its incorrect price (i.e., the noisy label) should depend on properties of the underlying stock. Such instance-dependent label noise is more challenging, as it may contain *spurious signals* that are easy to learn by certain architectures. In this section, we evaluate the representation' predictive power for three different GNNs trained with instance-dependent label noise.

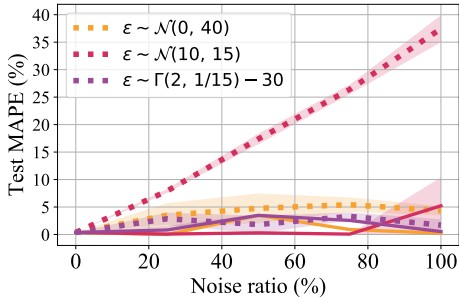

Figure 3: **Representations are very predictive for a GNN well-aligned with the target function under additive label noise.** On the maximum degree task, the representations' predictive power (solid lines) achieves low test MAPE ($< 5\%$) across all three types of noise for the max-sum GNN, despite that the model's test MAPE (dotted lines) may be quite large (for non-zero-mean noise). We average the statistics over 3 runs using different random seeds.

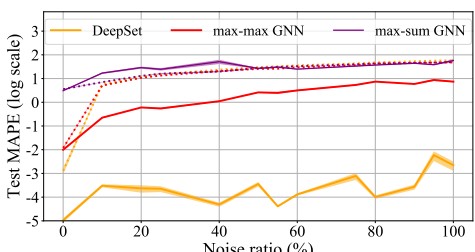

Figure 4: **Representations are more predictive for GNNs more aligned with the target function, and less predictive for GNNs more aligned with the noise function.** On the maximum node feature task, while all three GNNs have large test errors under high noise ratios (dotted lines), the predictive power (solid lines) in representations from Deepset (yellow) and max-max GNN (red) greatly reduces the test MAPE. In contrast, the representation's predictive power for max-sum GNN barely reduces the model's test MAPE (tiny gap between dotted and solid purple lines).

**Task and Label Noise.** We experiment with a new task—computing the maximum node feature:

$$f(\mathcal{G}) := \max_{u \in \mathcal{G}} ||\boldsymbol{x}_u||_\infty. \tag{10}$$

To create a instance-dependent noise, we randomly replace the label with the maximum degree:

$$g(\mathcal{G}) := \max_{u \in \mathcal{G}} \sum_{v \in \mathcal{N}(u)} 1. \tag{11}$$

**Architecture.** We consider three GNNs: DeepSet [83], max-max GNN, and max-sum GNN. DeepSet can be interpreted as a special GNN that does not use neighborhood information:

$$h_{\mathcal{G}} = \text{MLP}^{(1)}\Big(\max_{u \in \mathcal{G}} \text{MLP}^{(0)}\Big(\boldsymbol{x}_u\Big)\Big). \tag{12}$$

Max-max GNN is a 2-layer GNN with max-aggregation and max-readout. Max-sum GNN is the same as the one in the previous section.

DeepSet and max-max GNN are well-aligned with the target function $f(\mathcal{G})$, as their MLP modules only need to learn simple linear functions. In contrast, max-sum GNN is more aligned with $g(\mathcal{G})$ than $f(\mathcal{G})$ since neither its MLP modules or sum-aggregation module can efficiently learn the max-operation in $f(\mathcal{G})$ [55, 57].

Moreover, DeepSet cannot learn $g(\mathcal{G})$ as the model ignores *edge information*. We take the hidden representations before the last MLP modules in all three GNNs and compare their predictive power.

**Findings.** While all three GNNs have large test errors under high noise ratios (dotted lines in Figure 4), the predictive power in representations from GNNs more aligned with the target function — DeepSet (solid yellow line) and max-max GNN (solid red line) — significantly reduces the original models' test errors by 10 and 1000 times respectively. Yet, for the max-sum GNN, which is more aligned with the noise function, training with noisy labels indeed destroy the internal representations such that they are no longer to predict the target function — its representations' predictive power (solid purple line) barely decreases test error. We also evaluate the predictive power of these three types of randomly-initialized GNNs, and the results are in Table 4 (Appendix A.1).

## 4 Experiments on Vision Datasets

Many noisy label training methods are benchmarked on image classification; thus, we also validate our hypothesis on image domains. We compare the representations' predictive power between MLPs and CNN-based networks using 10% clean labels (all models are trained until they could perfectly fit the noisy labels, a.k.a., achieving close to 100% training accuracy). We further evaluate the predictive

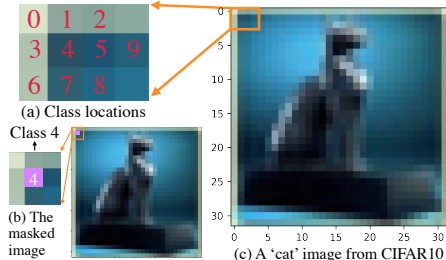

(a) Class locations

Class 4

(b) The masked image

(c) A 'cat' image from CIFAR10

Figure 5: **Synthetic Labels on CIFAR-Easy.** For each image, we mask a pixel at the top left corner with pink color. Then the synthetic label for this image is the location of the pink pixel/mask (i.e., the cat image in the above example has label 4).

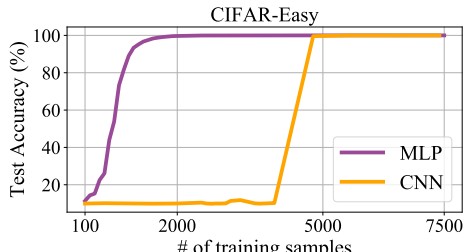

Figure 6: **Sample complexity of MLPs and CNNs on CIFAR-Easy.** Both MLPs and CNNs can achieve 100% test accuracy given sufficient training examples, but MLPs need far fewer examples than CNNs and thus are more sample-efficient on CIFAR-Easy.

power in representations learned with SOTA methods. Predictive power on networks that aligned well with the target function could further improve SOTA method's test performance (Section 4.2). The final model also outperforms some sophisticated methods on noisy label training which also use clean labels (Appendix A.4). All our experiment details are in Appendix B.4.

## 4.1 MLPs vs. CNN-based networks

To validate our hypothesis, we consider several target functions with different levels of alignments to MLPs and CNN-based networks. All models in this section are trained with standard procedures without any robust training methods or robust losses.

**Datasets and Label Noise.** We consider two types of target functions: one aligns better with CNN-based models than MLPs, and the other aligns better with MLPs than CNN-based networks.

**1). CIFAR-10** and **CIFAR-100** [84] come with clean labels. Therefore, we generate two types of noisy labels following existing works: (1) **uniform label noise** randomly replaces the true labels with all possible labels, and (2) **flipped label noise** swaps the labels between similar classes (e.g., deer↔horse, dog↔cat) on CIFAR-10 [49], or flips the labels to the next class on CIFAR-100 [8].

**2). CIFAR-Easy** is a dataset modified on CIFAR-10 with labels generated by procedures in Figure 5 — the class/label of each image depends on the location of a special pixel. We consider three types of noisy labels on CIFAR-Easy: (1) **uniform label noise** and (2) **flipped label noise** (described as above); and (3) **instance-dependent label noise** which takes the original image classification label as the noisy label.

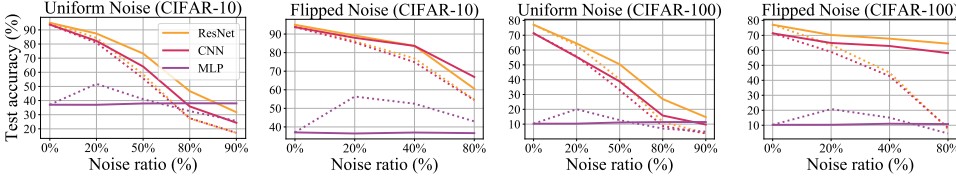

Figure 7: **CIFAR-10/100 with uniform and flipped label noise.** Each line indicates the raw test accuracy (dotted) and the predictive power in representations (solid) learned by a model trained across various noise ratios. As CNN-based networks align better with image classification tasks than MLPs, their representations' predictive power (solid yellow and red lines) are higher than that of MLPs (solid purple lines) on most noise ratios.

**Architectures.** On CIFAR-10/100, we evaluate the predictive power in representations for three architectures: 4-layer MLPs, 9-layer CNNs, and 18-layer PreAct ResNets [85]. On CIFAR-Easy, we compare between MLPs and CNNs. We take the representations before the penultimate layer when evaluating the predictive power for these networks.

As the designs of CNN-based networks (e.g., CNNs and ResNets) are similar to human perception system because of the receptive fields in convolutional layers and a hierarchical extraction of more

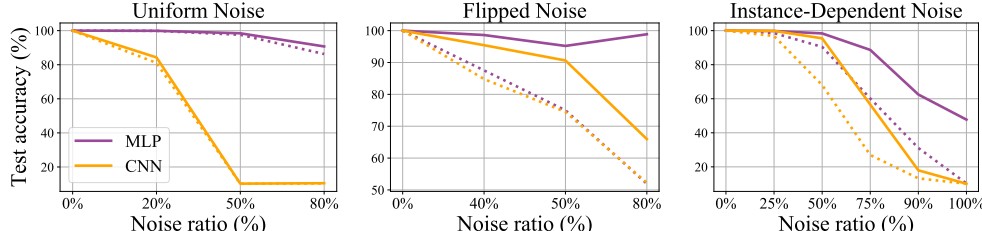

Figure 8: **CIFAR-Easy with uniform, flipped, and instance-dependent label noise.** Each line indicates the raw test accuracy (dotted) and the predictive power in representations (solid) learned by a model trained across various noise ratios. As MLPs align better with the target function than CNN-based networks on CIFAR-Easy, their representations' predictive power on MLPs (solid purple lines) are consistently better than that of CNNs (solid yellow lines) across various noise ratios and noise types.

Table 1: **Comparison of different networks' test accuracies (%) on CIFAR-10/100.** We color a test accuracy in red if it is lower than the test accuracy from vanilla training, and we color it in green if it is higher.

| Model | Setting | CIFAR10 | | | | | | | | CIFAR100 | | | | | | |
| | | Uniform noise | | | | Flipped noise | | | | Uniform noise | | | | Flipped noise | | |
| | | 20% | 50% | 80% | 90% | 20% | 40% | 80% | | 20% | 50% | 80% | 90% | 20% | 40% | 80% |
|---|---|---|---|---|---|---|---|---|---|---|---|---|---|---|---|---|
| 4-layer FC (MLP) | Vanilla training | 51.8 | 41.0 | 32.5 | 25.6 | 56.4 | 52.5 | 43.0 | | 20.1 | 12.7 | 7.0 | 4.9 | 20.8 | 15.1 | 4.5 |
| | DivideMix [49] | 62.2 | 55.2 | 34.4 | 28.1 | 60.2 | 56.8 | 44.0 | | 32.8 | 28.0 | 13.9 | 7.2 | 31.5 | 22.3 | 1.3 |
| | DivideMix's Predictive Power | 38.6 | 38.6 | 38.8 | 38.2 | 38.5 | 39.0 | 38.8 | | 11.1 | 11.8 | 12.4 | 11.6 | 11.1 | 12.0 | 11.9 |
| PreAct ResNet18 | Vanilla training | 84.4 | 58.5 | 27.3 | 17.2 | 86.1 | 76.9 | 54.7 | | 63.2 | 40.2 | 11.5 | 3.9 | 63.6 | 45.2 | 7.4 |
| | DivideMix [49] | 95.7 | 94.4 | 92.9 | 75.4 | 94.0 | 92.1 | 56.2 | | 76.9 | 74.2 | 59.6 | 31.0 | 77.0 | 55.2 | 0.2 |
| | DivideMix's Predictive Power | 96.0 | 94.8 | 93.5 | 83.8 | 94.9 | 94.0 | 93.6 | | 76.6 | 73.9 | 60.9 | 39.3 | 76.8 | 74.8 | 76.1 |
| 9-layer CNN | Vanilla training | 80.9 | 55.7 | 27.5 | 17.1 | 85.5 | 74.9 | 54.4 | | 55.8 | 33.1 | 8.8 | 3.7 | 59.0 | 42.8 | 8.3 |
| | DivideMix [49] | 94.5 | 93.4 | 91.2 | 78.2 | 92.9 | 89.8 | 55.3 | | 71.4 | 69.0 | 51.8 | 22.9 | 71.3 | 53.0 | 0.3 |
| | DivideMix's Predictive Power | 94.5 | 93.6 | 91.4 | 81.8 | 93.6 | 92.1 | 90.1 | | 69.9 | 67.1 | 50.4 | 26.3 | 70.2 | 69.0 | 68.8 |

and more abstracted features [86, 87], CNN-based networks are expected to *align better* with the target functions than MLPs on image classification datasets (e.g., CIFAR-10/100).

On the other hand, on CIFAR-Easy, while both CNNs and MLPs can generalize perfectly given sufficient training examples, MLPs have a much smaller sample complexity than CNNs (Figure 6). Thus, both MLP and CNN are *well-aligned* with the target function on CIFAR-Easy, but MLP is *better-aligned* than CNN according to Theorem 2. Moreover, since the instance-dependent label on CIFAR-Easy is the original image classification label, CNN is also *aligned* with this instance-dependent noise function on CIFAR-Easy.

**Experimental Results.** First, we empirically verify our hypothesis that *networks better-aligned with the target function have more predictive representations*. As expected, across most noise ratios on CIFAR-10/100, the representations in CNN-based networks (i.e., CNN and ResNet) are more predictive than those in MLPs (Figure 7) under both types of label noise. Moreover, the predictive power in representations learned by less aligned networks (i.e., MLPs) sometimes are even worse than the vanilla-trained models' test performance, suggesting that the noisy representations on less aligned networks may be more corrupted and less linearly separable. On the other hand, across all three types of label noise on CIFAR-Easy, MLPs, which align better with the target function, have more predictive representations than CNNs (Figure 8).

Table 2: **Comparison of different networks' test accuracies (%) on CIFAR-Easy.** We color a test accuracy in red if it is lower than the test accuracy from vanilla training, and we color it in green if it is higher.

| Model | Setting | Uniform noise | | | | Flipped noise | | | Spurious noise | | | | |
| | | 0% | 50% | 80% | 90% | 40% | 50% | 80% | 25% | 50% | 75% | 90% | 100% |
|---|---|---|---|---|---|---|---|---|---|---|---|---|---|
| 4-layer FC (MLP) | Vanilla training | 100.00 | 99.88 | 97.57 | 86.29 | 87.52 | 75.01 | 51.94 | 98.55 | 90.46 | 60.16 | 31.08 | 10.25 |
| | DivideMix [49] | 98.35 | 10.00* | 99.99 | 16.22 | 100.00 | 88.36 | 50.04 | 100.00 | 100.00 | 45.59 | 14.16 | 10.10 |
| | DivideMix's Predictive Power | 100.00 | 100.00 | 100.00 | 99.94 | 100.00 | 100.00 | 100.00 | 100.00 | 100.00 | 100.00 | 98.66 | 99.65 |
| 9-layer CNN | Vanilla training | 100.00 | 81.08 | 10.15 | 10.36 | 84.80 | 74.50 | 52.24 | 96.84 | 68.07 | 26.97 | 13.24 | 10.07 |
| | DivideMix [49] | 100.00 | 100.00 | 100.00 | 10.00 | 92.75 | 86.33 | 50.60 | 99.96 | 99.82 | 10.45 | 10.09 | 10.14 |
| | DivideMix's Predictive Power | 100.00 | 100.00 | 100.00 | 10.09 | 99.33 | 98.42 | 96.76 | 100.00 | 99.99 | 14.99 | 10.70 | 10.15 |

* The phenomenon that DivideMix fails under 50% uniform noise but succeeds under 80% uniform noise is due to the unstable behaviors of DivideMix's division process, indicating that the predictive power in representations could be a more stable measure of a model's robustness, as the model can fail miserably (a.k.a., performance close to random guessing), but its learned representation can still predict the target function well.

We also observe that *models with similar test performance could have various levels of predictive powers in their learned representations*. For example, in Figure 8, while the test accuracies of MLPs and CNNs are very similar on CIFAR-Easy under flipped label noise (i.e., dotted purple and yellow lines overlap), the predictive power in representations from MLPs is much stronger than the one from CNNs (i.e., solid purple lines are much higher than yellow lines). This also suggests that when trained with noisy labels, if we do not know which architecture is more aligned with the underlying target function, we can evaluate the predictive power in their representations to test alignment.

We further discover that *for networks well-aligned with the target function, its learned representations are more predictive when the noise function shares more mutual information with the target function*. We compute the empirical mutual information between the noisy training labels and the original clean labels across different noise ratios on various types of label noise. The predictive power in representations improves as the mutual information increases (Figure 11 in Appendix A). This explains why the predictive power for a network is often higher under flipped noise than uniform noise: at the same noise ratio, flipped noise has higher mutual information than uniform noise. Moreover, comparing across the three datasets in Figure 11, we observe the growth rate of a network's predictive power w.r.t. the mutual information depends on both the intrinsic difficulties of the learning task and the alignment between the network and the target function.

### 4.2   Predictive Power in Representations for Models Trained with SOTA Methods

As previous experiments are on standard training procedures, we also validate our hypothesis on models learned with SOTA methods on noisy label training. We evaluate the representations' predictive power for models trained with the SOTA method, DivideMix [49], which leverages techniques from semi-supervised learning to treat examples with unreliable labels as unlabeled data.

We compare (1) the test performance for models trained with standard procedures on noisy labels (denoted as **Vanilla training**), (2) the SOTA method's test performance (denoted as **DivideMix**), and (3) the predictive power in representations from models trained with DivideMix in (2) (denoted as **DivideMix's Predictive Power**).

We discover that *the effectiveness of DivideMix also depends on the alignment between the network and the target/noise functions*. DivideMix only slightly improves the test accuracy of MLPs on CIFAR-10/100 (Table 1), and DivideMix's predictive power does not improve the test performance of MLPs, either. In Table 2, DivideMix also barely helps CNNs as they are well-aligned with the instance-dependent noise, where the noisy label is the original image classification label.

Moreover, we observe that *even for networks well-aligned with the target function, DivideMix may only slightly improve or do not improve its test performance at all* (e.g., red entries of DivideMix on MLPs in Table 2). Yet, the representations learned with DivideMix can still be very predictive: the predictive power can achieve over 50% improvements over DivideMix for CNN-based models on CIFAR-10/100 (e.g., 80% flipped noise), and the improvements can be over 80% for MLPs on CIFAR-Easy (e.g., 90% uniform noise).

Tables 1 and 2 shows that the representations' predictive power on networks well aligned with the target function could further improve SOTA test performance. Appendix A.4 further demonstrates that on large-scale datasets with real-world noisy labels, the predictive power in well-aligned networks could outperform sophisticated methods that also use clean labels (Table 9 and Table 10).

## 5   Concluding Remarks

This paper is an initial step towards formally understanding how a network's architectures impacts its robustness to noisy labels. We formalize our intuitions and hypothesize that a network better-aligned with the target function would learn more predictive representations under noisy label training. We prove our hypothesis on a simplified noisy setting and conduct systematic experiments across various noisy settings to further validate our hypothesis.

Our empirical results along with Theorem 2 suggest that knowing more structures of the target function can help design more robust architectures. In practice, although an exact mathematical formula for a decomposition of a given target function is often hard to obtain, a high-level decomposition of the target function often exists for real-world tasks and will be helpful in designing robust architectures — a direction undervalued by existing works on learning with noisy labels.

## Acknowledgments

We thank Denny Wu, Xiaoyu Liu, Dongruo Zhou, Vedant Nanda, Ziyan Yang, Xiaoxiao Li, Jiahao Su, Wei Hu, Bo Han, Simon S. Du, Justin Brody, and Don Perlis for helpful feedback and insightful discussions. Additionally, we thank Houze Wang, Qin Yang, Xin Li, Guodong Zhang, Yixuan Ren, and Kai Wang for help with computing resources. This research is partially performed while Jingling Li is a remote research intern at the Vector Institute and the University of Toronto. Li and Dickerson were supported by an ARPA-E DIFFERENTIATE Award, NSF CAREER IIS-1846237, NSF CCF-1852352, NSF D-ISN #2039862, NIST MSE #20126334, NIH R01 NLM-013039-01, DARPA GARD #HR00112020007, DoD WHS #HQ003420F0035, and a Google Faculty Research Award. Ba is supported in part by the CIFAR AI Chairs program, LG Electronics, and NSERC. Xu is supported by NSF CAREER award 1553284 and NSF III 1900933. Xu is also partially supported by JST ERATO JPMJER1201 and JSPS Kakenhi JP18H05291. Zhang is supported by ODNI, IARPA, via the BETTER Program contract #2019-19051600005. The views and conclusions contained herein are those of the authors and should not be interpreted as necessarily representing the official policies, either expressed or implied, of ODNI, IARPA, or the U.S. Government. The U.S. Government is authorized to reproduce and distribute reprints for governmental purposes notwithstanding any copyright annotation therein.

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
