# Appendix: How does a Neural Network's Architecture Impact its Robustness to Noisy Labels?

## A    Additional Experimental Results

In this section, we include additional experimental results for the predictive power in (a) representations from randomly initialized models (Appendix A.1), (b) representations learned under different types off additive label noise (Appendix A.2) and (c) representations learned with a robust loss function (Appendix A.3). We further demonstrates that the predictive power in well-aligned networks could even outperform sophisticated methods that also utilize clean labels (Appendix A.4).

### A.1    Predictive Power of Randomly Initialized Models

We first evaluate the predictive power of randomly initialized models (a.k.a., untrained models), and we compare their results with GNNs trained on clean data (a.k.a., 0% noise ratio).

Table 3: Predictive power in representations from random and trained max-sum GNNs on the maximum degree task (Section 3.2). Notice that lower test MAPE denotes better test performance.

| Model | Test MAPE | |
|---|---|---|
| | Random | Trained |
| Max-sum GNN | $12.74 \pm 0.57$ | $0.37 \pm 0.08$ |

Table 4: Predictive power in representations from various types of random and trained GNNs on the maximum node feature task (Section 3.3). Notice that lower test MAPE denotes better test performance.

| Model | Test MAPE | | Test MAPE (log scale) | |
|---|---|---|---|---|
| | Random | Trained | Random | Trained |
| DeepSet | 5.14e-05 | 1.06e-05 | -4.29 | -4.97 |
| Max-max GNN | 0.794 | 0.0099 | -0.10 | -2.00 |
| Max-sum GNN | 54.28 | 3.08 | 1.73 | 0.488 |

### A.2    Additive Label Noise on Graph Algorithmic Datasets

We conduct additional experiments on additive label noise drawn from distributions with larger mean and larger variance. We consider four such distributions: Gaussian distributions $\mathcal{N}(10, 30)$ and $\mathcal{N}(20, 15)$, a long-tailed Gamma distribution with mean equal to 10: $\Gamma(2, \frac{1}{15}) - 20$, and another long-tailed t-distribution with mean equal to 10: $\mathcal{T}(\nu = 1) + 10$. Figure 9 demonstrates that for a GNN well aligned to the target function, its representations are still very predictive even under non-zero mean distributions with larger mean and large variance.

### A.3    Training with a Robust Loss Function

We also train the models with a robust loss function–Mean Absolute Error (MAE), and we observe similar trends in the representations' predictive power as training the models using MSE (Figure 10).

$$\text{loss} = \sum_{i=1}^{n} |y_{\text{true}} - y_{\text{pred}}|. \tag{13}$$

### A.4    Comparing with Sophisticated Methods Using Clean Labels

In previous experiments (section 4.2), we have shown that the predictive power in well-aligned models could further improve the test performance of SOTA methods on noisy label training. As we use a small set of clean labels to measure the predictive power, we also wonder how the improvements obtained by the predictive power compare with the sophisticated methods that also use clean labels.

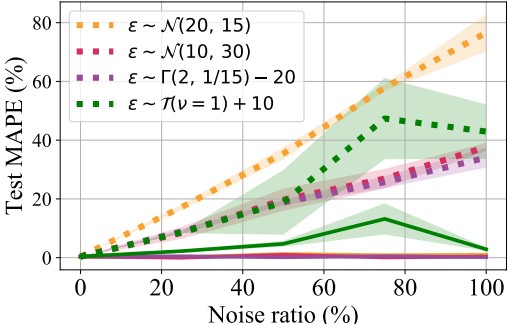

Figure 9: **Additional experiments on simple noise drawn from non-zero mean distributions**. On the maximum degree task, max-sum GNNs have large test errors (dotted lines) under additive label noise drawn from non-zero-mean distributions. Yet, the predictive power in representations (solid lines) greatly reduces the test errors with 10% clean labels.

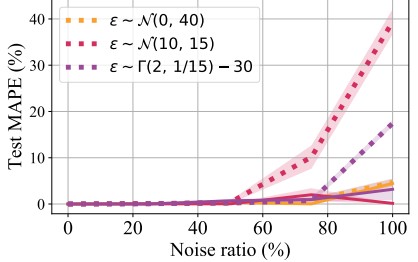

(a) Test errors of max-sum GNNs on the maximum degree task with additive label noise

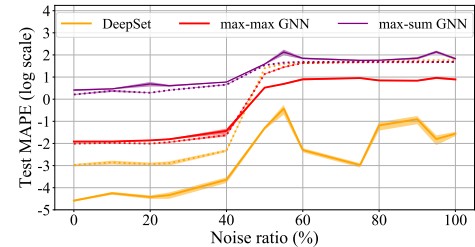

(b) Test errors of three different GNNs on the maximum node feature task with instance-dependent label noise

Figure 10: **Predictive power in representations trained with MAE**. For GNNs trained with MAE, the predictive power in representations exhibits similar trends as models trained with MSE. The robust loss function, MAE, is more helpful in learning more predictive representations under smaller noise ratios.

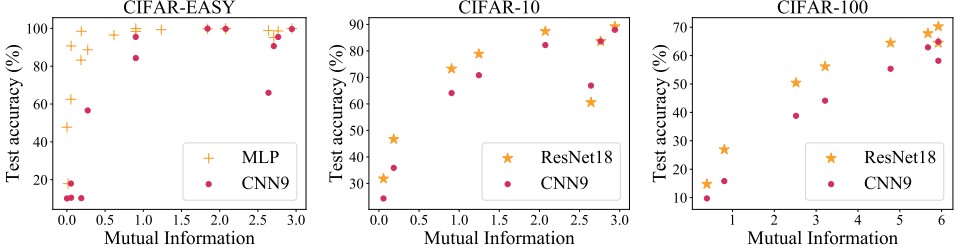

Figure 11: **The predictive in the representations grows as the mutual information between the noisy labels and original clean labels increases** for models well-aligned with the target function. The x-axis of each point in the plots denotes the mutual information between a given noisy dataset and the original clean labels. The corresponding y-axis denotes the representations' predictive power for a model trained on this noisy dataset with standard procedures (i.e., vanilla training).

### A.4.1 Sophisticated Methods Using Clean Labels

In our experiments, we consider the following methods which use clean labels: L2R [89], Mentor-Net [26], SELF [28], GLC [18], Meta-Weight-Net [90], and IEG [88]. Besides, as the SOTA method, DivideMix, keeps dividing the training data into labeled and unlabeled sets during training, we also compare with directly using clean labels in DivideMix: we mark the small set of clean data as labeled data during the semi-supervised learning step in DivideMix. We denote this method as *DivideMix w/ Clean Labels (DwC)* and further measure the predictive power in representations learned by DwC.

### A.4.2 Datasets

We conduct experiments on CIFAR-10/100 with synthetic noisy labels and on two large-scale datasets with real-world noisy labels: Clothing1M and Webvision.

Table 5: Test accuracy (%) on **CIFAR-10 with uniform label noise**.

| Method | # clean per class | Noise ratio | | | | | |
|---|---|---|---|---|---|---|---|
| | | 0% | 20% | 40% | 50% | 80% | 90% |
| Cross-Entropy | - | 95.0 ± 0.1 | 84.4 ± 0.4 | 67.9 ± 1.1 | 58.5 ± 1.5 | 27.3 ± 0.4 | 17.2 ± 0.5 |
| IEG [88] | 10 | 94.4 | 92.9 ± 0.2 | 92.5 ± 0.5 | - | 85.6 ± 1.1 | - |
| L2R [89] | 100 | 96.1 | 90.0 ± 0.4 | 86.9 ± 0.2 | - | 73.0 ± 0.8 | - |
| MentorNet [26] | 500 | 96.0 | 92.0 | 89.0 | - | 49.0 | - |
| DivideMix [49] | - | 95.0 ± 0.1 | 95.7 | 94.4 ± 0.1 | 94.4 | 92.9 | 75.4 |
| DivideMix w/ Clean Labels (DwC) | 500 | 95.0 ± 0.1 | 95.9 ± 0.1 | 94.3 ± 0.1 | 94.8 ± 0.1 | 92.9 ± 0.2 | 81.5 ± 0.4 |
| DivideMix's Predictive Power | 10 | 95.0 ± 0.1 | **96.0 ± 0.1** | **94.5 ± 0.1** | 94.7 ± 0.0 | **93.2 ± 0.1** | 74.6 ± 0.4 |
| DivideMix's Predictive Power | 100 | 95.0 ± 0.1 | **96.1 ± 0.1** | **94.5 ± 0.1** | 94.8 ± 0.1 | **93.4 ± 0.1** | 76.7 ± 0.4 |
| DivideMix's Predictive Power | 500 | 95.0 ± 0.1 | **96.1 ± 0.1** | **94.6 ± 0.1** | **94.9 ± 0.1** | **93.6 ± 0.1** | 80.4 ± 0.4 |
| DwC's Predictive Power | 500 | 95.0 ± 0.1 | **95.9 ± 0.1** | **94.7 ± 0.1** | **95.0 ± 0.1** | **93.5 ± 0.1** | **87.4 ± 0.2** |

Table 6: Test accuracy (%) on **CIFAR-10 with flipped label noise**.

| Method | # clean per class | Noise ratio | | |
|---|---|---|---|---|
| | | 20% | 40% | 80% |
| Cross-Entropy | - | 86.1 ± 0.5 | 76.9 ± 1.0 | 54.7 ± 0.7 |
| IEG [88] | 10 | 92.7 ± 0.2 | 90.2 ± 0.5 | 78.9 ± 3.5 |
| SELF [28] | 100 | 92.8 | 89.1 | - |
| GLC [18] | 100 | 89.7 ± 0.3 | 88.9 ± 0.2 | - |
| Meta-Weight-Net [90] | 100 | 90.3 ± 0.6 | 87.5 ± 0.2 | - |
| DivideMix [49] | - | 94.0 ± 0.3 | 92.1 | 56.2 ± 0.1 |
| DivideMix w/ Clean Labels (DwC) | 500 | 94.2 ± 0.2 | 91.7 ± 0.3 | 56.9 ± 0.4 |
| DivideMix's Predictive Power | 10 | 93.68 ± 0.34 | **92.14 ± 0.50** | **88.71 ± 0.46** |
| DivideMix's Predictive Power | 100 | **94.63 ± 0.09** | **93.59 ± 0.12** | **92.68 ± 0.11** |
| DivideMix's Predictive Power | 500 | **95.00 ± 0.10** | **94.25 ± 0.11** | **93.87 ± 0.09** |
| DwC's Predictive Power | 500 | **94.89 ± 0.11** | **93.53 ± 0.12** | **92.84 ± 0.12** |

**Clothing1M** [58] has real-world noisy labels with an estimated 38.5% noise ratio. The dataset has a small human-verified training data, which we use as clean data. Following recent method [49], we use 1000 mini-batches in each epoch to train models on Clothing1M.

**WebVision** [93] also has real-world noisy labels with an estimated 20% noise ratio. It shares the same 1000 classes as ImageNet [94]. For a fair comparison, we follow [26] to create a mini WebVision dataset with the top 50 classes from the Google image subset of WebVision. We train all models on mini WebVision dataset and evaluate on both the WebVision and ImageNet validation sets. We select 100 images per class from ImageNet training data as clean data.

### A.4.3 Experimental Settings

We use the same architectures and hyperparameters as DivideMix: an 18-layer PreAct Resnet [85] for CIFAR-10/100, a ResNet-50 pre-trained on ImageNet for Clothing1M, and Inception-ResNet-

Table 7: Test accuracy (%) on **CIFAR-100 with uniform label noise**.

| Method | # clean per class | Noise ratio | | | | | |
|---|---|---|---|---|---|---|---|
| | | 0% | 20% | 40% | 50% | 80% | 90% |
| Cross-Entropy | - | 77.1 ± 0.1 | 63.2 ± 0.2 | 49.8 ± 0.3 | 40.2 ± 0.2 | 11.5 ± 0.1 | 3.9 ± 0.1 |
| IEG [88] | 10 | 72.1 | 69.3 ± 0.5 | 67.0 ± 0.8 | - | 60.7 ± 1.0 | - |
| L2R [89] | 10 | 81.2 | 67.1 ± 0.1 | 61.3 ± 2.0 | - | 35.1 ± 1.2 | - |
| MentorNet [26] | 50 | 79.0 | 73.0 | 68.0 | - | 35.0 | - |
| DivideMix [49] | - | 77.1 ± 0.1 | 76.9 | 74.8 ± 0.2 | 74.2 | 59.6 | 31.0 |
| DivideMix w/ Clean Labels (DwC) | 50 | 77.1 ± 0.1 | 76.8 ± 0.2 | 75.0 ± 0.2 | 74.0 ± 0.2 | 60.4 ± 0.2 | 39.8 ± 0.1 |
| DivideMix's Predictive Power | 10 | 77.1 ± 0.1 | 76.3 ± 0.2 | 74.0 ± 0.1 | 73.6 ± 0.2 | 58.5 ± 0.2 | 32.6 ± 0.4 |
| DivideMix's Predictive Power | 50 | 77.1 ± 0.1 | **77.2 ± 0.2** | **75.1 ± 0.1** | **74.7 ± 0.2** | **61.1 ± 0.1** | 37.6 ± 0.3 |
| DwC's Predictive Power | 50 | 77.1 ± 0.1 | 76.4 ± 0.2 | 74.6 ± 0.1 | 73.7 ± 0.2 | **61.5 ± 0.1** | **45.1 ± 0.2** |

Table 8: Test accuracy (%) on **CIFAR-100 with flipped label noise**.

| Method | # clean per class | Noise ratio | | |
|---|---|---|---|---|
| | | 20% | 40% | 80% |
| Cross-Entropy | - | $63.6 \pm 0.5$ | $45.2 \pm 0.3$ | $7.4 \pm 0.2$ |
| GLC [18] | 10 | $63.1 \pm 0.5$ | $62.2 \pm 0.6$ | - |
| Meta-Weight-Net [90] | 10 | $64.2 \pm 0.3$ | $58.6 \pm 0.5$ | - |
| DivideMix [49] | - | $77.0 \pm 0.2$ | $55.2 \pm 0.7$ | $0.2 \pm 0.0$ |
| DivideMix w/ Clean Labels (DwC) | 50 | $76.9 \pm 0.2$ | $55.4 \pm 0.8$ | $0.2 \pm 0.0$ |
| DivideMix's Predictive Power | 10 | $74.31 \pm 0.16$ | $\mathbf{72.09 \pm 0.24}$ | $\mathbf{73.75 \pm 0.31}$ |
| DivideMix's Predictive Power | 50 | $76.74 \pm 0.18$ | $\mathbf{74.91 \pm 0.22}$ | $\mathbf{76.13 \pm 0.21}$ |
| DwC's Predictive Power | 50 | $76.35 \pm 0.18$ | $\mathbf{74.46 \pm 0.23}$ | $\mathbf{75.55 \pm 0.22}$ |

Table 9: Comparison with state-of-the-art methods in test accuracy (%) on Clothing1M.

| Method | # clean | Test Accuracy |
|---|---|---|
| Cross-Entropy | - | 69.21 |
| DivideMix [49] | - | 74.76 |
| IEG [88] | 50k | 77.21 |
| CleanNet [91] | 50k | 79.9 |
| F-correction [20] | 50k | 80.38 |
| Self-learning [92] | 50k | **81.16** |
| DivideMix+Ours | 50k | **80.47** |

V2 [95] for WebVision. We use the test accuracy reported in the original papers whenever possible, and the accuracy for L2R [89] are from [88]. For IEG, we use the reported test accuracy obtained by ResNet-29 rather than WRN28-10, because ResNet-29 has a comparable number of parameters as the PreAct ResNet-18 we use.

As CIFAR-10/100 do not have a validation set, we follow previous works to report the averaged test accuracy over the last 10 epochs: we measure the predictive power in representations for models from these epochs and report the averaged test accuracy. For Clothing1M and Webvision, we use the associated validation set to select the best model and measure the predictive power in its representations.

### A.4.4   Results

Tables 5-8 show the results on CIFAR-10 and CIFAR-100 with uniform and flipped label noise, where **boldfaced numbers** denote test accuracies better than all methods we compared with. We see that across different noise ratios on CIFAR-10/100 with flipped label noise, the predictive power in representations remains roughly the same as the test performance of the model trained on clean data for a network well-aligned with the target function, which matches with Lemma 4. For CIFAR-10 with uniform label noise, the predictive power in representations achieves better test performance using only 10 clean labels per class on most noise ratios; for CIFAR-100 with uniform label noise, the predictive power in representations could achieve better test performance using only 50 labels per class.

Moreover, we observe that adding clean data to the labeled set in DivideMix (DwC) may barely improve the model's test performance when the noise ratio is small and under flipped label noise. At 90% uniform label noise, DwC can greatly improve the model's test performance, and the predictive power in representations can achieve a even higher test accuracy with the same set of clean data used to train DwC.

On Clothing1M, we compare the predictive power in representations learned by DivideMix with existing methods that use the small set of human-verified data: CleanNet [91], F-correction [20] and Self-learning [92]. As these methods also use the clean subset to fine-tune the whole model, we follow similar procedures to fine-tune the model (trained by DivideMix) for 10 epochs and then select the best model based on the validation accuracy to measure the predictive power in its representations.

Table 10: Comparison with state-of-the-art methods trained on (mini) WebVision dataset. Numbers denote top-1 (top-5) accuracy (%) on the WebVision and the ImageNet validation sets.

| Method | WebVision | | ILSVRC12 | |
|---|---|---|---|---|
| | top1 | top5 | top1 | top5 |
| MentorNet [26] | 63.00 | 81.40 | 57.80 | 79.92 |
| IEG [88] | - | - | **80.0** | **94.9** |
| DivideMix [49] | 77.32 | 91.64 | 75.20 | 90.84 |
| DivideMix+Ours | **77.70 ± 0.23** | 90.68 | 75.99 ± 0.09 | 91.30 |

The predictive power in representations could further improve the test accuracy of DivideMix by around 6% and outperform IEG, CleanNet, and F-correction (Table 9). The improved test accuracy is also competitive to [92], which uses a much more complicated learning framework.

On Webvision, the predictive power also improves the model's test performance (Table 10). The improvement is less significant than on Clothing1M as the estimated noise ratio on Webvision (20%) is smaller than Clothing1M (38.5%).

## B  Experimental Details

### B.1  Computing Resources

We conduct all the experiments on one NVIDIA RTX 2080 Ti GPU, except for the experiment on the WebVision dataset [93] (Table 10), which uses 4 GPUs concurrently.

### B.2  Measuring the Predictive Power

We use linear regression to train the linear model when measuring the predictive power in representations. For representations from all models except MLPs, we use ordinary least squares linear regression (OLS). When the learned representations are from MLPs, we e use ridge regression with penalty = 1 since we find the linear models trained by OLS may easily overfit to the small set of clean labels.

### B.3  Experimental Details on GNNs

**Common settings.**    In the generated datasets, each graph $\mathcal{G}$ is sampled from Erdős-Rényi random graphs with an edge probability uniformly chosen from $\{0.1, 0.2, \cdots, 0.9\}$. This sampling procedure generates diverse graph structures. The training and validation sets contain 10,000 and 2,000 graphs respectively, and the number of nodes in each graph is randomly picked from $\{20, 21, \cdots, 40\}$. The test set contains 10,000 graphs, and the number of nodes in each graph is randomly picked from $\{50, 51, \cdots, 70\}$.

#### B.3.1  Additive Label Noise

**Dataset Details.**    In each graph, the node feature $\boldsymbol{x}_u$ is a scalar randomly drawn from $\{1, 2, \cdots, 100\}$ for all $u \in \mathcal{G}$.

**Model and hyperparameter settings.**    We consider a 2-layer GNN with max-aggregation and sum-readout (max-sum GNN):

$$\boldsymbol{h}_G = \text{MLP}^{(2)}\Big(\max_{u \in \mathcal{G}} \sum_{v \in \mathcal{N}(u)} \text{MLP}^{(1)}\Big(\boldsymbol{h}_u, \boldsymbol{h}_v\Big)\Big), \boldsymbol{h}_u = \sum_{v \in \mathcal{N}(u)} \text{MLP}^{(0)}\Big(\boldsymbol{x}_u, \boldsymbol{x}_v\Big).$$

The width of all MLP modules are set to 128. The number of layers are set to 3 for $\text{MLP}^{(0)}$ and $\text{MLP}^{(1)}$. The number of layers are set to 1 for $\text{MLP}^{(2)}$. We train the max-sum GNNs with loss function MSE or MAE for 200 epochs. We use the Adam optimizer with default parameters, zero weight decay, and initial learning rate set to 0.001. The batch size is set to 64. We early-stop based on a noisy validation set.

### B.3.2 Instance-Dependent Label Noise.

**Dataset Details.**  Since the task is to predict the maximum node feature and we use the maximum degree as the noisy label, the correlation between true labels and noisy labels are very high on large and dense graphs if the node features are uniformly sampled from $\{1, 2, \cdots, 100\}$. To avoid this, we use a two-step method to sample the node features. For each graph $\mathcal{G}$, we first sample a constant upper-bound $M_{\mathcal{G}}$ uniformly from $\{20, 21, \cdots, 100\}$. For each node $u \in \mathcal{G}$, the node feature $\boldsymbol{x}_u$ is then drawn from $\{1, 2, \cdots, M_{\mathcal{G}}\}$.

**Model and hyperparameter settings.**  We consider a 2-layer GNN with max-aggregation and sum-readout (max-sum GNN), a 2-layer GNN with max-aggregation and max-readout (max-max GNN), and a special GNN (DeepSet) that does not use edge information:

$$h_G = \text{MLP}^{(1)}\Big(\max_{u \in \mathcal{G}} \text{MLP}^{(0)}\big(\boldsymbol{x}_u\big)\Big).$$

The width of all MLP modules are set to $128$. The number of layers is set to 3 for $\text{MLP}^{(0)}, \text{MLP}^{(1)}$ in max-max and max-sum GNNs and for $\text{MLP}^{(0)}$ in DeepSet. The number of layers is set to 1 for $\text{MLP}^{(2)}$ in max-max and max-sum GNNs and for $\text{MLP}^{(1)}$ in DeepSet. We train these GNNs with MSE or MAE as the loss function for 600 epochs. We use the Adam optimizer with zero weight decay. We set the initial learning rate to $0.005$ for DeepSet and $0.001$ for max-max GNNs and max-sum GNNs. The models are selected from the last epoch so that they can overfit the noisy labels more.

## B.4 Experimental Details on Vision Datasets

**Neural Network Architectures.**  Table 11 describes the 9-layer CNN [96] used on CIFAR-Easy and CIFAR-10/100, which contains 9 convolutional layers and 19 trainable layers in total. Table 12 describes the 4-layer MLP used on CIFAR-Easy and CIFAR-10/100, which has 4 linear layers and ReLU as the activation function.

Table 11: 9-layer CNN on CIFAR-Easy and CIFAR-10/100.

| Input | 32×32 Color Image |
|---|---|
| Block 1 | Conv(3×3, 128)-BN-LReLU
Conv(3×3, 128)-BN-LReLU
Conv(3×3, 128)-BN-LReLU
MaxPool(2×2, stride = 2)
Dropout(p = 0.25) |
| Block 2 | Conv(3×3, 256)-BN-LReLU
Conv(3×3, 256)-BN-LReLU
Conv(3×3, 256)-BN-LReLU
MaxPool(2×2, stride = 2)
Dropout(p = 0.25) |
| Block 3 | Conv(3×3, 512)-BN-LReLU
Conv(3×3, 256)-BN-LReLU
Conv(3×3, 128)-BN-LReLU
GlobalAvgPool(128) |
| Score | Linear(128, 10 or 100) |

Table 12: 4-layer FC on CIFAR-Easy and CIFAR-10/100.

| Input | 32×32 Color Image |
|---|---|
| Block 1 | Linear(32×32×3, 512)-ReLU
Linear(512, 512)-ReLU
Linear(512, 512-ReLU |
| Score | Linear(512, 10 or 100) |

**Vanilla Training.**  For models trained with standard procedures, we use SGD with a momentum of $0.9$, a weight decay of $0.0005$, and a batch size of $128$. For ResNets and CNNs, the initial learning rate is set to $0.1$ on CIFAR-10/100 and $0.01$ on CIFAR-Easy. For MLPs, the initial learning rate is set to $0.01$ on CIFAR-10/100 and $0.001$ on CIFAR-Easy. The initial learning rate is multiplied by $0.99$ per epoch on CIFAR-10/100, and it is decayed by 10 after 150 and 225 epochs on CIFAR-Easy.

**Train Models with SOTA Methods.**  We use the same set of hyperparameter settings from DivideMix [49] to obtain corresponding trained models and measure the predictive power in representations from these models.

On CIFAR-10/100 with flipped noise, we only use the small set of clean labels to train the linear model in our method, and the clean subset is randomly selected from the training data. On CIFAR-10/100 with uniform noise, the clean labels we use are from examples with highest model uncertainty [97]. Besides the clean set, we also use randomly-sampled training examples labeled with the model's original predictions to train the linear model. We use 5,000 such samples under 20%, 40%, 50%, and 80% noise ratios, and we use 500 such samples under 90% noise ratio.

## C   Theoretical Results

We first provide a formal version of Theorem 1 based on [55]. Theorem 1 connects a network's architectural alignment with the target function to its learned representations' predictive power *when trained on clean data.*

**Theorem 3.** *(Better alignment implies better predictive power on clean training data; [55]). Fix $\epsilon$ and $\delta$. Given a target function $f : \mathcal{X} \to \mathcal{Y}$ that can be decomposed into functions $f_1, ..., f_n$ and given a network $\mathcal{N}$, where $\mathcal{N}_1, ..., \mathcal{N}_n$ are $\mathcal{N}$'s modules in sequential order, suppose the training dataset $\mathcal{S} := \{\boldsymbol{x}_j, y_j\}_{j=1}^{M}$ contains $M$ i.i.d. samples drawn from a distribution with clean labels $y_j := f(x_j)$. Then under the following assumptions, Alignment$(\mathcal{N}, f, \epsilon, \delta) \leq M$ if and only if there exists a learning algorithm $A$ such that the network's last module $\mathcal{N}_n$'s representations learned by $A$ on the training data $\mathcal{S}$ have predictive power $P_n(f, \mathcal{N}, \mathcal{S}) \leq \epsilon$ with probability $1 - \delta$.*

*Assumptions:*

*(a) We train each module $\mathcal{N}_i$'s sequentially: for each $\mathcal{N}_i$, the input samples are $\{h^{(i-1)}(\boldsymbol{x}_j), f_i(h^{(i-1)}(\boldsymbol{x}_j))\}_{j=1}^{M}$ with $h^{(0)}(\boldsymbol{x}) = \boldsymbol{x}$. Notice that each input $h^{(i-1)}(\boldsymbol{x}_j)$ is the output from the previous modules, but its label is generated by the function $f_i$ on $h^{(i-1)}(\boldsymbol{x}_j)$.*

*(b) For the clean training set $\mathcal{S}$, let $\mathcal{S}' := \{\hat{\boldsymbol{x}}_j, y_j\}_{j=1}^{M}$ denote the perturbed training data ($\hat{\boldsymbol{x}}_j$ and $\boldsymbol{x}_j$ share the same label $y_j$). Let $f_{\mathcal{N},A}$ and $f'_{\mathcal{N},A}$ denote the functions obtained by the learning algorithm $A$ operating on $\mathcal{S}$ and $\mathcal{S}'$ respectively. Then for any $\boldsymbol{x} \in \mathcal{X}$, $\|f_{\mathcal{N},A}(\boldsymbol{x}) - f'_{\mathcal{N},A}(\boldsymbol{x})\| \leq L_0 \cdot \max_{\boldsymbol{x}_j \in \mathcal{S}} \|\boldsymbol{x}_j - \hat{\boldsymbol{x}}_j\|$, for some constant $L_0$.*

*(c) For each module $\mathcal{N}_i$, let $\hat{f}_i$ denotes its corresponding function learned by the algorithm $A$. Then for any $\boldsymbol{x}, \hat{\boldsymbol{x}} \in \mathcal{X}$, $\|\hat{f}_j(\boldsymbol{x}) - \hat{f}_j(\hat{\boldsymbol{x}})\| \leq L_1 \|\boldsymbol{x} - \hat{\boldsymbol{x}}\|$, for some constant $L_1$.*

We have empirically shown that Theorem 3 also hold when we train the models on noisy data. Meanwhile, we prove Theorem 3 for a simplified noisy setting where the target function and noise function share a common feature space, but have different prediction rules. For example, the target function and noise function share the same feature space under flipped label noise (in classification setting). Yet, their mappings from the learned features to the associated labels are different.

**Theorem 4.** *(Better alignment implies better predictive power on noisy training data). Fix $\epsilon$ and $\delta$. Let $\{\boldsymbol{x}_j\}_{j=1}^{M}$ be i.i.d. samples drawn from a distribution. Given a target function $f : \mathcal{X} \to \mathcal{Y}$ and a noise function $g : \mathcal{X} \to \mathcal{Y}$, let $y := f(\boldsymbol{x})$ denote the true label for an input $\boldsymbol{x}$, and $\hat{y} := g(\boldsymbol{x})$ denote the noisy label of $\boldsymbol{x}$. Let $\hat{\mathcal{S}} := \{(\boldsymbol{x}_j, y_j)\}_{j=1}^{N} \bigcup \{(\boldsymbol{x}_j, \hat{y}_j)\}_{j=N+1}^{M}$ denote a noisy training set with $M - N$ noisy samples for some $N \in \{1, 2, \cdots, M\}$. Given a network $\mathcal{N}$ with modules $\mathcal{N}_i$, suppose $\mathcal{N}$ is well-aligned with the target function $f$ (i.e., the alignment between $\mathcal{N}$ and $f$ is less than $M$ — Alignment$(\mathcal{N}, f, \epsilon, \delta) \leq M$). Then under the same assumptions in Theorem 3 and the additional assumptions below, there exists a learning algorithm $A$ and a module $\mathcal{N}_i$ such that when training the network $\mathcal{N}$ on the noisy data $\hat{\mathcal{S}}$ with algorithm $A$, the representations from its $i$-th module have predictive power $P_i(f, \mathcal{N}, \mathcal{C}) \leq \epsilon$ with probability $1 - \delta$, where $\mathcal{C}$ is a small set of clean data with a size greater than the number of dimensions in the output of module $\mathcal{N}_i$.*

*Additional assumptions (a simplified noisy setting):*

*(a) There exists a function $h$ on the input domain $\mathcal{X}$ such that the target function $f : \mathcal{X} \to \mathcal{Y}$ and the noise function $g : \mathcal{X} \to \mathcal{Y}$ can be decomposed as: $f(\boldsymbol{x}) = f_r(h(\boldsymbol{x}))$ with $f_r$ being a linear function and $g(\boldsymbol{x}) = g_r(h(\boldsymbol{x}))$ for some function $g_r$.*
*(b) $f_r$ is a linear map from a high-dimensional space to a low-dimensional space.*
*(c) The loss function used in measuring the predictive power is mean squared error (denoted as $\| \cdot \|$).*

**Remark.** Theorem 4 suggests that the representations' predictive power for models well aligned with the target function should remain roughly similar across different noise ratios under flipped label noise. Empirically, we observe similar phenomenons in Figures 7-8, and in Tables 6 and 8. Some discrepancy between the experimental and theoretical results could exist under vanilla training as Theorem 4 assumes sequential training, which is different from standard training procedures.

**Proof of Theorem 4.** According to the definition of alignment in Definition 2, since $Alignment(\mathcal{N}, f, \epsilon, \delta) \leq M$ and $f(\boldsymbol{x}) = f_r(h(\boldsymbol{x}))$, we can find a sub-structure (denoted as $\mathcal{N}_{sub}$) in the network $\mathcal{N}$ with sequential modules $\{\mathcal{N}_1, \cdots, \mathcal{N}_i\}$ such that $\mathcal{N}_{sub}$ can efficiently learn the function $h$ (i.e., the sample complexity for $\mathcal{N}_{sub}$ to learn $h$ is no larger than $M$). According to Theorem 3, applying sequential learning to train $\mathcal{N}_{sub}$ with labels $h(\boldsymbol{x})$, the representations of $\mathcal{N}_{sub}$ will have predictive power $P_i(h, \mathcal{N}_{sub}, \mathcal{C}) \leq \epsilon$ with probability $1 - \delta$.

Since for each input $\boldsymbol{x}$ in the noisy training data $\hat{\mathcal{S}}$, its label can be written as $f_r(h(\boldsymbol{x}))$ (if it is clean) or $g_r(h(\boldsymbol{x}))$ (if it is noisy), when the network $\mathcal{N}$ is trained on $\hat{\mathcal{S}}$ using sequential learning, its sub-structure $\mathcal{N}_{sub}$ can still learn $h$ efficiently (i.e., $\mathcal{M}_A(h, \mathcal{N}_{sub}, \epsilon, \delta) \leq M$ for some learning algorithm $A$). Thus, the representations learned from the noisy training data $\hat{\mathcal{S}}$ can still be very predictive (i.e., $P_i(h, \mathcal{N}_{sub}, \mathcal{C}) \leq \epsilon$ with probability $1 - \delta$).

Since $f_r$ is a linear map from a high-dimensional space to a low-dimensional space, and the clean data $\mathcal{C}$ has enough samples to learn $f_r$ ($|\mathcal{C}|$ is larger than the input dimension of $f_r$), the linear model $L$ learned by linear regression can also generalize $f_r$ (since linear regression has a closed form solution in this case as the problem is over-complete). Therefore, as $P_i(h, \mathcal{N}_{sub}, \mathcal{C}) \leq \epsilon$, $P_i(f, \mathcal{N}_{sub}, \mathcal{C}) \leq \epsilon$ also holds. Notice that $P_i(f, \mathcal{N}_{sub}, \mathcal{C}) = P_i(f, \mathcal{N}, \mathcal{C})$ as $\mathcal{N}_i$ is also the $i$-th module in $\mathcal{N}$. Hence, we have shown that there exist some module $\mathcal{N}_i$ such that $P_i(f, \mathcal{N}, \mathcal{C}) \leq \epsilon$ with probability $1 - \delta$.