# OpenReview forum: "How does a Neural Network's Architecture Impact its Robustness to Noisy Labels?"
_NeurIPS.cc/2021/Conference — NeurIPS 2021 Poster_

### Official Review · Reviewer_RRS5 · 2021-07-08

**Rating:** 5
**Confidence:** 2

**Summary:**

The key contribution of the paper is a notation of alignment between the architecture of the network and the target function. The paper develops theoretical analysis and a set of empirical experiments to demonstrate that networks with better alignment can learn better representations even with noisy labels.

**Limitations And Societal Impact:**

The authors have limited descriptions about the limitation of the paper. By moving some graphs into the appendix the author may be able to add a separated paragraph in the main paper to discuss the limitation of their work.

**Main Review:**

I find the paper’s idea very interesting and can be potentially very useful in more than one areas, i.e. network architecture search, training with noisy data, learning better representations. However, I am having more and more questions as I read through the paper (details to follow) and I am happy to increase my score if the authors can provide more explanations to resolve my concerns.

### Originality
This work provides a bridge between an existing measurement, alignment [1], and Predictive Power [2]. The related work is well-cited.


### Quality
The motivation of this work is intuitive and easy to understand but the analysis and the result in the current submission does not clearly explain (this could also be a problem in writing and I will clarify in the next paragraph) to me why these analytical and empirical results support the claim: *networks with better alignment have more predictive power*. I have following questions regarding both theorems  and the findings from the experiments:
- About Theorem 1. A clear proof of Theorem 1 (or  Theorem 3 in the appendix) and its connection with [1] is missing. The author claims that Theorem 3 (from appendix) is a formal version of Theorem 3 and cites [1] as their proofs. However, it lacks a formal proof to elaborate more in formal math expressions in order to clarify which and how definitions and theorems in [1] are used to derive Theorem 3.

- About Theorem 2. Is there a typo in the line 155 -156? Why “a network  N well-aligned with f” is equivalent to $P_j(f, N, C)$ is small as $P_j(f, N, C) < c$ is previously defined as predictive power instead of the alignment. Are there missing assumptions about C in Theorem 2? For example, there is no assumption stated in Theorem 2 about C but in the proof of Theorem 4, the formal version of Theorem 2, the authors further assume $|C|$ is larger than the input dimensions of $f_r$ (line 773). If  $|C|$ is necessary to prove the theorem then it would be necessary to provide the corresponding condition when the theorem will hold.

- Is there a way tp empirically calculate the alignment score instead of a qualitative measurement based on assumptions? Statistical tests and even some graphs are convincing when validating a quantity A (alignment) is positively correlated with another quantity B (predictive power). However, it seems that alignment is not able to be empirically computed even for the simplest GNN case. The author instead provides qualitative descriptions of the alignment scores for all target networks evaluated in this paper, e.g. “max-sum GNN is more aligned”, “CNN-based models are expected to align better than MLPs”. These descriptions are not rigorously justified either by theorems or empirically justified by calculating the alignment score with its definition given in this paper. It would help to convince that there is indeed a positive correlation between the alignment and predictive power if the alignment can be quantitatively measured and evaluated.

### Clarity
There are a few suggestions for improving the writing of this paper.
- I am having a hard time separating which part of the analysis in section 2 is from a prior work (mostly from [1] and [2]) and which part is novel and should be considered as the main contributions of this paper. I would suggest separating the prior work and the paper’s work in a more obvious way.

- I am not sure I understand the PCA experiment and its analysis. Can the authors explain more about the expected outcome of PCA and why it supports the claim made at line 210.

### Significance
The result of this paper can be very useful and potentially significant for people to search network architecture in training noisy dataset if the paper can provide more clear theoretical and empirical justifications.


[1] Xu, K., Li, J., Zhang, M., Du, S., Kawarabayashi, K., & Jegelka, S. (2020). What Can Neural Networks Reason About? ICLR 2020.

[2] Alain, G., & Bengio, Y. (2017). Understanding intermediate layers using linear classifier probes. ArXiv, abs/1610.01644. ICLR 2017.


**Time Spent Reviewing:**

12

---

> ### Author Response · Authors · 2021-08-14
> **Reply to Reviewer RRS5: Thank you for the detailed review. We hope you could reconsider your evaluation.**
>
> We appreciate the reviewer's feedback, and we would like to clarify some of the reviewer's concerns and questions, as discussed below.
>
> > How could the theorems in [1] derive Theorem 3?
>
> Thanks for asking. Theorem 3 is derived from Theorem 3.6 in [1] where we use the same set of symbols to denote the modules in a neural network (a.k.a., $\mathcal{N}_1, \cdots, \mathcal{N}_n$) and the decomposition of the target function (a.k.a., $f_1, \cdots, f_n$). The notation “$(M, \epsilon, \delta)$-learnable” in Theorem 3.6 [1] is defined in Definition 3.3 in [1], and it corresponds to the statement "$P_n(f, \mathcal{N}, \mathcal{S}) \leq \epsilon$ with probability $1 - \delta$" in our Theorem 3. Here, $M$ denotes the size of the training data $\mathcal{S}$. The assumptions in Theorem 3.6 [1] and Theorem 3 are also the same.
>
> Theorem 3.6 [1] proves the “only if” direction in Theorem 3: lower (better) predictive implies lower (better) alignment measure. The “if” direction in Theorem 3 is trivial as we can just set $n=1$ (a.k.a., the network $\mathcal{N}$ only has one module, $\mathcal{N}_1 = \mathcal{N}$, and $f_1(\cdot) = f(\cdot)$ for the target function). Then if the predictive power $P_1(f, \mathcal{N}, \mathcal{S}) \leq \epsilon$ with probability $1 - \delta,$ the sample complexity for the network $\mathcal{N}$ to learn the target function $f$ would be bounded by $M$ because:
>
> $\mathcal{M}_{A_1} (f, \mathcal{N}, \epsilon, \delta) = \mathcal{M} (f_1, \mathcal{N}_1, \epsilon, \delta) \leq |S| = M$,
>
> which gives us $\text{Alignment}(\mathcal{N}, f, \epsilon, \delta) < M$ by Definition 2.
>
> We will include the above explanations and proof sketch in our revision.
>
> > “Why “a network N well-aligned with f” is equivalent to $P_j(f, \mathcal{N}, \mathcal{C})$ is small as $P_j(f, \mathcal{N}, \mathcal{C})$ is previously defined as predictive power instead of the alignment.”
>
> We would like to draw the reviewer’s attention to Theorem 1 (lines 129-132), which states that a network well-aligned with the target function will have a small (better) predictive power (a.k.a, if $\text{Alignment}(\mathcal{N}, f, \epsilon, \delta) < |C|$, then $P_j(f, \mathcal{N}, \mathcal{C}) < \epsilon$ with probability $1 - \delta$ for all $j \in \\{1, \cdots, n\\}$). Thus, in lines 155-156, we are given a set of clean data $|C|$ and a small constant $c$, and we suppose the network $\mathcal{N}$ is well aligned with the target function $f$ (a.k.a., $\text{Alignment}(\mathcal{N}, f, c, \delta) < |C|$). Thus, it is natural that we will have
> $P_j(f, \mathcal{N}, \mathcal{C}) < c$.
>
> > Adding additional assumption on the size of the clean data $C$ to Theorem 2 and 4.
>
> Thanks for the suggestion. We need |C| to be larger than the input dimensions of $f_r$ to obtain a closed solution in linear regression. We have stated this assumption explicitly in Theorem 4 (lines 748-749), and we will add this assumption to Theorem 2 in our updated version.
>
> > Is there a way to empirically calculate the alignment score instead of a qualitative measurement based on assumptions?
>
> Yes. We can empirically estimate the alignment between a network and a given target function by looking at the sample complexity for this network to learn the given target function. This empirical measure is equivalent to setting $n=1$ in Definition 2 (the definition of the alignment measure).
>
> Moreover, we would like to draw the reviewer’s attention to Figure 6, where we provide a quantitative comparison between the alignments of CNN and MLP with the target function on CIFAR-Easy. For our GNN experiments, as prior works [1, 3] have provided extensive theoretical analyses on the alignments of various GNNs, we provide qualitative comparisons and refer readers to these previous works.
>
> > “I would suggest separating the prior work and the paper’s work in a more obvious way.”
>
> Thanks for the suggestion. We would like to clarify that although the techniques we introduced in this work are mainly based on prior work [1, 2], our contribution is not the techniques we introduced but (1) our discovery that despite having large test errors, networks well-aligned with the target function can still be robust to noisy labels when evaluating their predictive power in representations; (2) our theoretical and empirical justifications for the finding in (1), and (3) our improvements over SOTA noisy-label-training methods utilizing this finding.
>
> One of our main theoretical contributions is extending Theorem 1 (Theorem 3.6 in [1]) to noisy label settings. At a high level, Theorem 1 states that a network well-aligned with the target function can achieve good predictive power when trained on clean data, and our Theorem 2 shows that a network well-aligned with the target function can also achieve good predictive power when trained on noisy data. Though Theorem 2 is on a simplified noisy label setting, its proof is non-trivial. Also, this simplified noisy setting is practical in the real world (e.g., additive label noise and instance-dependent label noise both belong to this setting).
>
> We will make our contributions clearer from previous work in our updated version.
>
> > The PCA experiment and its analysis: “Can the authors explain more about the expected outcome of PCA and why it supports the claim made at line 210.”
>
> Thanks for pointing this out. We will include more context in the caption of Figure 2: the x-axis and y-axis of Figure 2 denote the largest and the 2nd largest principal component directions of the PCA on the learned representations from a max-sum GNN (rather than MLPs) obtained under a 100% noise ratio. Each dot in Figure 2 denotes a single training example, and the dot is colored by the true label. The linear relationship between the representations and the true label is demonstrated by the gradual change of colors from left to right, indicating that the true label correlates well with the largest principal component of the learned representation.
>
> This phenomenon supports the claim made at line 210 as the representations learned under noisy labels can be mapped into a compact space (e.g., the space constructed by the principal components) that has a linear correlation with the true label, and such mapping, as well as the linear correlation, could be recovered by linear regression.
>
> [1] Xu, K., Li, J., Zhang, M., Du, S., Kawarabayashi, K., & Jegelka, S. (2020). What Can Neural Networks Reason About? ICLR 2020.
>
> [2] Alain, G., & Bengio, Y. (2017). Understanding intermediate layers using linear classifier probes. ArXiv, abs/1610.01644. ICLR 2017.
>
> [3] Xu, K., Zhang, M., Li, J., Du, S., Kawarabayashi, K., & Jegelka, S. (2021). How Neural Networks Extrapolate: From Feedforward to Graph Neural Networks. ICLR 2021.
>
> ---
>
> We hope that our responses so far have cleared up the confusion for the reviewer to reevaluate and to discuss our contribution. We are happy to provide more context in the follow-up discussion. Please let us know if there is anything else we could clarify.

---

### Official Review · Reviewer_AauR · 2021-07-16

**Rating:** 8
**Confidence:** 2

**Summary:**

The paper studies robustness to noisy labels in the context of network architectures. It provides theoretical framework to study predictive powers of the representations trained with noisy labels, and defines alignment between network and target function using sample complexity to learn the task. The authors hypothesize that the representations learned by networks that are better aligned to for the target function, have higher predictive powers. They demonstrate this both theoretically and empirically.


**Limitations And Societal Impact:**

While the checklist mentions that these are discussed in the supplementary materials, I could not find a section/untitled paragraphs dedicated to this.

**Main Review:**

This is a very well-written paper with valuable theoretical and empirical contributions.

The paper formalizes the concepts of predictive power and alignment of the network for a target task. Both of these are modular i.e., enables studying the network at intermediate levels/modules too. This setup helped showcase that the learned representations can be predictive despite being trained on noisy labels.

The authors hypothesize that the networks with lower sample complexity/better alignment to model the target function, learn representations with higher predictive power. This is a very intuitive statement: if the design/mechanisms used within the architecture are better aligned with the task, then it already has an inductive bias to model the task well. Of course, the interesting part here is that this is true even when the training samples are noisy. Theorem 2 formalizes this and the empirical results on graph networks and vision tasks provide empirical support to this claim.

Overall, this is a clearly written paper and the setup used here will be useful for future studies on network robustness.

I have some comments/suggestions:

[C1] Sec 3.2: High predictive power at 100% noise level is especially surprising. Untrained/Randomly initialized model would have been a good sanity check i.e., how much predictive power can be attributed to the architecture alone?  This would have been interesting for sec 3.3 too, to check if the architecture-alignment can be solely attributed to the architecture itself.

[C2] It would have been very interesting to compare vision transformers against CNNs. A recent paper [1] shows that vision transformers are more akin to human vision in terms of the errors they make. So, this would be a relevant study here.

[C3] Based on the formulations provided in the paper, lower values of predictive power (P) and Alignment(.) are better, but since they have positive connotations in “real life” one would expect higher values to be better. It makes things a bit confusing while reading.


[1] Tuli, Shikhar, et al. "Are Convolutional Neural Networks or Transformers more like human vision?." arXiv preprint arXiv:2105.07197 (2021).



**Time Spent Reviewing:**

4

---

> ### Author Response · Authors · 2021-08-14
> **Reply to Reviewer AauR: Thank you for the thoughtful review.**
>
> Thank you for appreciating our work and providing insightful feedback! We have added additional results based on your questions and have addressed the technical comments below:
>
> > Evaluating the predictive power of randomly initialized models
>
> Thanks for the suggestion. We have added the results for models that are randomly initialized, and we also compare them to the results of GNNs trained on clean data (0% noise ratio):
>
>   * Max degree task (Figure 3): predictive power of random and trained max-sum GNNs
> (lower test MAPE denotes better performance):
>
>     * | Model      | max-sum GNN (random) | max-sum GNN (trained) |
> |------------|----------------------|-----------------------|
> | Test MAPE  | 12.74 +/- 0.57       | 0.37 +/- 0.08         |
>
>   * Max node feature task (Figure 4): predictive power of various types of random and trained GNNs  (lower test MAPE denotes better performance):
>
>     * | Model                          | DeepSet  (random) | DeepSet  (trained) | max-max GNN  (random) | max-max GNN  (trained) | max-sum GNN  (random) | max-sum GNN  (trained) |
> |--------------------------------|-------------------|--------------------|-----------------------|------------------------|-----------------------|------------------------|
> | Test MAPE                      | 5.14e-05          | 1.06e-05           | 0.794                 | 0.0099                 | 54.28                 | 3.08                   |
> | Test MAPE (log scale, base 10) | -4.29             | -4.97              | -0.10                 | -2.00                  | 1.73                  | 0.488                  |
>
> We will include the above results in our updated version.
>
> ---
> Thank you for the suggestions in comment 2 and comment 3. We think comment 2 would be an interesting experiment to add for future work. For comment 3,  to avoid possible confusion, we will emphasize ahead in our revision that lower values of predictive power (P) and Alignment(.) are better (e.g., like test errors, lower values are often better).

---

### Official Review · Reviewer_mMWd · 2021-07-16

**Rating:** 6
**Confidence:** 3

**Summary:**

This work studied the connection between the neural network's architecture and its robustness to noisy labels. The architecture is quantified by its alignment with the target function, and the robustness is quantified by the predictive power of the learned representations. The authors propose a practical framework to compute the alignment and predictive power and hypothesize that the better alignment with the clean target function implies better predictive power and hence better robustness to noisy labels. They further conducted carefully designed experiments whose results support their hypothesis.

**Limitations And Societal Impact:**

Limitations and societal impact are addressed.

**Main Review:**

This paper is overall clear and well-written.

pros :
1. in the study of learning with noisy labels, people often assume the expressiveness of the underlying neural network model and focus on the robust loss function designation or transition matrix estimation. This work addresses the effect of the neural network architecture concerning its comparability with the training data. It demonstrates that the predictive power of the learned representations relates to the network architecture in a non-trivial way.
2. the authors proposed a practical framework to quantitatively measure the representation's predictive power and the alignment of the network with the training data.
3. the authors conducted carefully designed experiments on synthetic tasks as well as real-world image classification tasks. The results support their hypothesis: better alignment implies better predictive power. They also provided proof of the hypothesis under a simplified scenario.

cons:
1. one important aspect of the alignment concept in this paper is that it not only measures the capacity of learning but also measures the easiness of learning. This should have some relation to the spectral bias of neural network models. Such spectral bias refers to a common belief that the neural network models tend to learn "simple" patterns first before overfitting to the noise during training. It would be better if the authors could discuss the connection between these two concepts.
2. it is well-known that normally neural network is over-parameterized and can perfectly fit the noisy data during training. I wonder if the learned representations still possess good predictive power after the model is trained to perfectly overfit the noisy training dataset. I haven't found a description of whether the models achieve 100% training accuracy in the experiments.

**Time Spent Reviewing:**

5

---

> ### Author Response · Authors · 2021-08-14
> **Reply to Reviewer mMWd: Thank you for providing constructive feedback.**
>
> Thank you for appreciating our work and providing helpful feedback! We will include more discussions based on the questions you've asked, and we have addressed your detailed comments below:
>
> > How does the alignment connect to the spectral bias of a neural network?
>
> This is a great question. The spectral bias can be viewed as a special case of our proposed alignment measure: while the spectral bias denotes the simplicity bias of a network, the alignment measure defines what is considered as “simple” to learn for a given neural network (e.g., different architectures could regard different functions as the ones simple to learn). We will discuss this connection more in our revision.
>
> >  “I wonder if the learned representations still possess good predictive power after the model is trained to perfectly overfit the noisy training dataset.”
>
> This is a good point. We indeed train the models till convergence before reporting the predictive power in representations. On classification settings, most times, both MLPs and CNNs could perfectly fit the noisy labels (a.k.a., close to 100% training accuracy). On regression settings, for the task with the additive label noise, the training MAPE is still kind of large even when the GNN already converges since the GNN model needs to have a very large capacity to fit random Gaussian noise (e.g., the number of hidden units should grow with the number of training samples). For the task with instance-dependent label noise, max-max GNN and max-sum GNN can achieve a relatively low training MAPE but not DeepSet as DeepSet does not use the neighborhood information in its message-passing step, which makes it hard to learn/memorize the noisy labels.

---

### Official Review · Reviewer_NWJ7 · 2021-07-16

**Rating:** 5
**Confidence:** 3

**Summary:**

This paper investigates an interesting problem on how the network architecture is related to its performance under noisy labels. The paper offers a thorough foundation on the related topics and offers a good empirical discussion to support its main findings. Overall, it could be an interesting contribution to the community.

**Ethical Concerns:**

none noted.

**Limitations And Societal Impact:**

yes.

**Main Review:**

The paper offers a line of formal discussion and empirical investigation on the relationship between the alignment of the model architecture to the target functions and the robustness to the noisy labels. The overall methodology is interesting and the contributions seem solid, my major concern is that the main conclusion of the paper on "a network is more robust to noisy labels if its architecture is more aligned with the target function than the noise" seems quite intuitive; and the result on vision discusses CNN vs MLP seems a low-hanging fruit. Other than this major concern, I have several technical concerns.

1. The results do not seem to necessarily indicate the relationship between "the alignment of the model architecture to the target functions and the robustness to the noisy labels", in contrast, in many places, it seems only about "the alignment of the model architecture to the target functions and *test accuracy*". For example, in Figure 7, I don't see additional information if we accept the fact that CNN and ResNet simply learn the target function better (independent of noise level).

2. In Figure 8, I'm surprised to see that CNN suffers this much from the easy construction of the label, maybe some details of the model configurations and training procedures will be helpful to further review the results.
     * My conjecture is that while we can see CNN as a over-complicated model for this scenario, the overcomplicateness does not seem matter that much statistically speaking. However, whether CNN will align with the noise label better might matter much and offer an explanation to the performance gap between CNN and MLP. However, even in this argument, the way the authors construct the noisy labels do not seem to suggest any intentionally alignment between the noises and the labels.

3. Results in Table 3 does not seem a fair comparison because the vanilla performances of CNNs are much higher than the ones from MLP. Maybe reporting the increment will be better to present the main point.

4. Some writings of the paper seem overconfident.
     * One issue is is that while the authors indicate that their finding is that the alignment between architecture and target function is more important than the alignment between architecture and the noise function. It seems that there are no explicit discussions/experiments between the alignment and the noise functions, at least at the vision section.
     * I was very excited when I read the first half of the paper because the alignment towards a target function is an open-research question nowadays, and a hard one. But later, the authors only use MLP vs CNN to simulate whether the alignment is done well or not seems an under-delivery of what's promised in the first half of the paper.

**Time Spent Reviewing:**

3.5

---

> ### Author Response · Authors · 2021-08-14
> **Reply to Reviewer NWJ7: Thank you for the detailed review. We hope you could reconsider your evaluation.**
>
> We appreciate the reviewer's feedback. We would like to clarify some of the reviewer's concerns and questions, as discussed below.
>
> > First, we would like to clarify that our main finding --- networks more aligned to the target function would learn more robust representations --- is beyond common intuition.
>
> Previous works on learning with noisy labels often regard networks with large test errors as not resilient to noisy labels. Yet, we discover that despite having large test errors, networks well-aligned with the target function can still be robust to noisy labels when evaluating their predictive power in representations. Moreover, though works on learning with noisy labels have investigated many factors that may affect a network’s robustness, such as optimization, early-stopping, loss function, and so on, to the best of our knowledge, we are the first work that connects a network’s robustness to noisy labels with the inductive bias in its architecture.
>
> > "The results do not seem to necessarily indicate the relationship between 'the alignment of the model architecture to the target functions and the robustness to the noisy labels,' in contrast, in many places, it seems only about 'the alignment of the model architecture to the target functions and test accuracy'".
>
> As mentioned above, previous works often use a network’s raw test performance after training with noisy labels to indicate its robustness, which regards models having large test errors but useful representations as not robust to noisy labels. Though the definition of our predictive power can also be viewed as a measure of test errors, it is evaluated on the representations from modules in the network rather than from the whole network. The idea behind this definition is that an architecture with good performance on clean data often contains modules that align well with the target function. If these modules are not aligned with the noise function or if they are only aligned well with the common structures shared between the target and noise functions, the representations from these modules are likely to still contain useful information to predict the target function.
>
> Therefore, another takeaway from Figure 7 is that models with similar test performance could have various levels of predictive powers in their learned representations: on large noise ratios, ResNet and CNN have similar test accuracies (dotted yellow and red lines overlap), but they still have varying levels of predictive power (solid yellow and red lines still have gaps).
>
> > Details of model configurations and training procedures to explain the large performance drop of CNN on CIFAR-Easy (Figure 8)
>
> We would like to draw the reviewer’s attention to Appendix B, which contains the training details for all our experiments and model configurations (e.g., Table 9 and 10). In our work, we also provide some possible reasons for the larger performance drop of CNN:
>
> 1. As shown in Figure 6, CNN has a much larger sample complexity than MLP, which means that it would need more clean samples to achieve a good test accuracy.
>
> 2. As demonstrated in Figure 5, the way we construct the synthetic label on CIFAR-Easy is meant to be hard for the convolutional filters to learn it: the synthetic label of an image is dependent only on a small region of pixels (3*3 + 1) and is very sensitive to the exact location of the special pixel (thus not space-invariant). As the filter size in the CNN we choose is 3-by-3, learning the location of the special pixel could be a hard task for this CNN.
>
> 3. The instance-dependent label noise is also designed such that CNN can learn it more easily. The large performance drop may also be due to the good alignment between CNN and the noise function, as the reviewer has mentioned.
>
> > Missing discussions/experiments between the alignment and the noise functions
>
> We would like to draw the reviewer’s attention to section 3.3, where we explicitly construct an instance-dependent noise label such that (a) the max-sum GNN is aligned with this noise function, and (b) DeepSet cannot learn the noise function well as the model ignores edge information, which is used to calcluate the noise function.
>
> > “[T]he authors only use MLP vs. CNN to simulate whether the alignment is done well or not seems an under-delivery of what's promised in the first half of the paper.”
>
> Our experiments are beyond MLP vs. CNN. We would like to draw the reviewer’s attention to section 3, where we validate our hypothesis on synthetic graph algorithmic tasks by designing GNNs with different levels of alignments to the underlying target/noise functions. We conduct these experiments in section 3 since the theoretical properties of GNNs and their alignment with algorithmic regression tasks are well-studied [44–46, 61].
>
> ---
>
> We hope that our responses so far have cleared up the confusion for the reviewer to reevaluate and to discuss our contribution. We are happy to provide more context in the follow-up discussion. Please let us know if there is anything else we could clarify.

---

> > ### Comment · Reviewer_NWJ7 · 2021-08-19
> > **Continued Discussion**
> >
> > Thank you for the detailed responses to my concerns. Unfortunately, I do not consider several of the discussions convincing enough. For examples,
> >
> > 1. I would be reluctant to agree that "we are the first work that connects a network’s robustness to noisy labels with the inductive bias in its architecture". Depending on what the authors consider as "noisy labels", I would say that given the relationship between "noisy labels vs. features" and "labels vs. noisy features", I would say the main argument is intuitive. For example, [1] and many follow-ups aimed to explain the observation in [1], such as [2] from the theoretical perspective and [3] for the empirical perspective. If the authors do not agree with the relationship between "noisy labels vs. features" and "labels vs. noisy features", it might be OK to say that they are the first work.
> >
> > 2. The synthetic experiment in Section 3, in my opinion, seems easier to construct than the ones in Section 4, some further clarifications could be helpful.
> >
> > 3. It may not be the best strategy that the reviewer only quote half of my questions to respond, for example, my question was " It seems that there are no explicit discussions/experiments between the alignment and the noise functions, at least at the vision section", which suggests question to the vision section, and the authors quote and respond to the first half of the question by offering evidence that they had a discussion in other parts.
> >
> > [1] Understanding deep learning requires rethinking generalization
> >
> > [2] Regularization Matters: Generalization and Optimization of Neural Nets v.s. their Induced Kernel
> >
> > [3] High Frequency Component Helps Explain the Generalization of Convolutional Neural Networks

---

> > > ### Author Response · Authors · 2021-08-20
> > > **Reply to continued discussion**
> > >
> > > Thanks for the response. We respectfully disagree with the unconvincing points the reviewer listed:
> > >
> > > 1. We would like to address that [1], as discussed in our related work section, only studies one type of noisy labels under classification setting -- the uniform label noise (it also studies noise in the input, but that's beyond the scope of our work). Yet, in real-world scenarios, label noise is often data-dependent or at least label-dependent (the noisy label correlates with the original label) rather than randomly distributed. Our work not only extends the study of [1] to a diverse set of noisy labels (including ones that are more close to noisy labels), but also explains different affects brought by noisy label training from a architectural perspective, whereas the study of [2] is more on regularization technique, and the study of [3] is more on robustness to adversarial examples, both of which differ from our work in terms of the focus and problem settings.
> > >
> > > 2. We respectfully disagree with the reviewer's statement that "The synthetic experiment in Section 3, in my opinion, is even easier to construct than the ones in Section 4, thus I can hardly consider the response valid." First, it is unfair to treat supporting experimental results as invalid simply because the experiments are easier to construct. Second, the choices of target and noise function as well as setting the data distributions in Section 3 are non-trivial (e.g., Dataset Details in Appendix B.3). Moreover, as the reviewer is interested in the topic "the alignment towards a target function", many current works on this topic focus on GNN's alignments with algorithmic functions [4-7], and our experimental results in Section 3 explicitly support our main hypothesis.
> > >
> > > 3. We would like to draw the reviewer's attention to our response that the "instance-dependent label noise (in our vision experiment (CIFAR_EASY) is also designed such that CNN can learn it more easily". To elaborate on this, as stated in our paper (line 256-257) the instance-dependent label noise is the CIFAR-10 label, which can be regarded as a noise function that aligns better with CNNs than with MLPs. We are happy to provide more context if the reviewer has further questions on this point. Moreover, we would like to draw the reviewer's attention to the [NeurIPS reviewer guidelines](https://neurips.cc/Conferences/2021/Reviewer-Guidelines) for a more thoughtful, fair and respectful discussion.
> > >
> > > [1] Zhang, Chiyuan, et al. "Understanding deep learning (still) requires rethinking generalization." Communications of the ACM 64.3 (2021): 107-115.
> > >
> > > [2] Wei, Colin, et al. "Regularization matters: Generalization and optimization of neural nets vs their induced kernel." (2019).
> > >
> > > [3] Wang, Haohan, et al. "High-frequency component helps explain the generalization of convolutional neural networks." Proceedings of the IEEE/CVF Conference on Computer Vision and Pattern Recognition. 2020.
> > >
> > > [4] Xu, Keyulu, et al. "What can neural networks reason about?" In International Conference on Learning Representations, 2020.
> > >
> > > [5] Du, Simon, et al. "Graph neural tangent kernel: Fusing graph neural networks with graph kernels." In Advances in Neural Information Processing Systems, pages 5724–5734, 2019.
> > >
> > > [6] Xu, Keyulu, et al. "How neural networks extrapolate: From feedforward to graph neural networks." In International Conference on Learning Representations, 2021.
> > >
> > > [7] Velickovic, Petar, et al. "Neural execution of graph algorithms." In International Conference on Learning Representations, 2020.

---

### Official Review · Reviewer_8N9C · 2021-07-25

**Rating:** 6
**Confidence:** 5

**Summary:**

This paper focuses on the connection between the neural network’s architecture and its robustness when learning with noisy labels. The authors provide both theoretical and experimental results to justify their claims.

**Limitations And Societal Impact:**

The core idea and theorem rely on strong assumptions, but the explanations are not enough.

**Main Review:**

Learning with noisy labels is one of the hottest topics in weakly supervised learning. This paper gives new insights into this topic. More specifically, the authors first borrow the tool named the predictive power which shows that how good the learned representations are at predicting the target function. Then the definition of alignment is introduced, and the connection between predictive power and alignment is built to describe the model robustness. The authors provide some theoretical analyses and extensive experimental results to support their claims.

**Strength**
- The motivation of this paper is clear. Prior works mainly focus on designing robust algorithms from the perspectives of loss/label correction and sample selection, etc. This paper aims to investigate the influence of the network’s architecture on model robustness. Such a perspective is novel, and the core idea is well motivated.
- The experimental results are appreciated. To support their statements, the authors give detailed experimental results to show that better alignment could imply better robustness, and good representations could be learned even though there are noisy labels. The experiments are comprehensive, which include the MLP, GNN, and CNN architecture. Various types of label noise, e.g., additive, class-dependent, and instance-dependent label noise are also considered.

**Weakness \& Question**
- The writing of this paper is not clear. Also, it is not very logical. It is a bit hard to understand this paper. The authors give too many notations and definitions in this paper. However, the core idea lacks enough descriptions at a high level. Also, the results are extensive. But the following analyses and discussions are not enough.
- The definition of the alignment relies on the measurement of sample complexity. In this paper, how to measure it under deep networks? I notice Hypothesis 1 in Section 2.4. However, in Appendix, I found the formal version of Theorem 1 and the following analyses. Where are the details for justifying Hypothesis 1? Besides, more explanations for rationality are needed to be added. At the present stage, it seems that the authors only directly transfer the statements on clean data to the statements on noisy data.
- Figure 2 is not very intuitive. The authors state that the representations have a clear linear relationship with true labels. More explanations are expected to be provided. In addition, the irrelevance between the raw test accuracy and the predictive power when using MLP is needed to be further justified.
- The results on the 4-layer MLP in Table 2 are confusing. Why DevideMix can work well in the case of 80\% uniform noise, but fail in the less challenging case, i.e.,  50\% uniform noise?
- The issues/concerns of proofs in Appendix C. \
(1) Although Theorem 3 is borrowed from [1], it would be better if three assumptions can be further explained. [1] Keyulu Xu et al. What can neural networks reason about? In ICLR, 2020. \
(2) The main theorem of this paper, i.e., Theorem 4 in Appendix C, is a bit weak. Expect three assumptions in Theorem 3, additional three assumptions are added. However, the rationality of these assumptions is not provided. The assumptions (b) and (c) are relatively intuitive. The assumption (a) is hard to understand. Why the models trained on clean data and noisy data would have the same $h(x)$. If the $f_r$ is regarded as an FC layer, $h(x)$ means the representations? And why they will be the same for the target function and noise function? It is somewhat confusing for me about this assumption. \
(3) As clean labels still are diagonally dominant in noisy labels, it is reasonable that deep networks trained noisy data still be very predictive. In the proof of Theorem 4, the authors state that ''we can ﬁnd a sub-structure (denoted as $\mathcal{N}_{sub}$ in the network $\mathcal{N}$ with sequential modules'' to learn the function $h$. However, definition 2 is used on clean data before. Is it an assumption on noisy data? If not, please justify it and add more discussions.

**Time Spent Reviewing:**

10

---

> ### Author Response · Authors · 2021-08-14
> **Reply to Reviewer 8N9C: Thank you for the detailed review. We hope you could reconsider your evaluation.**
>
> We appreciate the reviewer's feedback and will add more explanations to better convey the high-level idea of our main finding and our theoretical results in the revision. We would like to clarify some of the reviewer's concerns and questions, as discussed below.
>
> > “The core idea lacks enough descriptions at a high level.”
>
> The main novelty of our work is not our proposed concepts/techniques, but our finding (as well as our theoretical and empirical results to support this finding) that despite having large test errors, networks well-aligned with the target function can still be robust to noisy labels when evaluating the predictive power in their learned representations. This finding has been addressed in our abstract, introduction, and conclusion (e.g., lines 3-8, 323-327), and we will make this finding clearer in our revision.
>
> > How to measure the alignment on deep networks?
>
> In practice, if the target function is obtuse/does not have a structural decomposition or if we don’t know the structures of the target function, we can still obtain an empirical estimation of the alignment measure by setting n=1 in definition 2 and directly measure the sample complexity of the network learning the given target function. Figure 6 is an example of how we use this way to measure and compare the MLPs’ and CNNs’ alignments to the target function in CIFAR-Easy.
>
> > “Where are the details for justifying Hypothesis 1?”  + Justification of assumption (a) in Theorem 4.
>
> We would like to draw the reviewer’s attention to the proof of Theorem 4 in appendix C, which contains the theoretical justifications for hypothesis 1 on a simplified noise setting. A high-level justification for assumption (a) in Theorem 4 is that in real-world data, the noisy label often has some dependency on the true label (e.g., additive label noise and flipped label noise); thus, the noise function would often share some commonalities with the target function, and we denote these commonalities as the function $h(\cdot)$. Then if some modules in a network are aligned with $h(\cdot)$, the network could still learn representations predictive to the target function when trained with noisy labels. We will make the above intuitions clearer in our revision.
>
> > “Figure 2 is not very intuitive...In addition, the irrelevance between the raw test accuracy and the predictive power when using MLP is needed to be further justified.”
>
> Thanks for the suggestion. We will include more context in the caption of Figure 2: the x-axis and y-axis of Figure 2 denote the largest and the 2nd largest principal component directions of the PCA on the learned representations from a max-sum GNN (rather than MLPs) obtained under a 100% noise ratio. Each dot in Figure 2 denotes a single training example, and the dot is colored by the true label. The linear relationship between the representations and the true label is demonstrated by the gradual change of colors from left to right, indicating that the true label correlates well with the largest principal component of the learned representation. This phenomenon is surprising as the max-sum GNN is learned under a 100% noise ratio and has a large test error (the test MAPE is around 35 according to Figure 3).
>
> > “The results on the 4-layer MLP in Table 2 are confusing. Why DevideMix can work well in the case of 80% uniform noise, but fail in the less challenging case, i.e., 50% uniform noise?”
>
> This is a great point. The phenomenon the reviewer pointed out is because of one disadvantage of the method DivideMix [38]: DivideMix, as its name suggest, divide the data into two sets (labeled and unlabeled), and we observe this division process is very unstable, especially when applying DivideMix on CIFAR-Easy --- the division can be very skewed (e.g., having almost 100% data in one set and nearly no data in the other set) and is very sensitive to choices of hyperparameters (e.g., random seed, model size, and so on). Such skewed division of the training data could lead to failures of the optimization step, and that’s why you see a 10% test accuracy (a.k.a. equivalent to random guess) for the phenomenon in table 2. The raw test accuracy of DivideMix varies greatly when we choose hyperparameters differently (e.g., using a different random seed or increasing the number of layers in MLP could lead to a test accuracy close to 100%).
>
> Yet, these failures and unstable behaviors of DivideMix actually support our contribution: the predictive power in representations could be a more stable and better measure of a model’s robustness, as the model can fail miserably (a.k.a., performance close to random guessing), but its learned representation can still predict the target function well.
>
> > Rationality of assumptions in Theorem 1 and 4
>
> In high level, Theorem 1 (Theorem 3.6 in [44]) states that networks more aligned with the target function learn more predictive representations on clean data, and Theorem 4 (our main theorem) shows that this also holds on a simplified noisy label setting. The three assumptions in Theorem 1, sequential learning, algorithmic stability, and Lipschitzness, are standard assumptions for obtaining theoretical results under the PAC learning framework [a-d].
>
> For assumption (a) in Theorem 4, $h(x)$ denotes the commonalities shared between the target function $f(x)$ and the noise function $g(x)$. At high-level, we can treat $h(\cdot)$ as the common features used in both target and noise functions. For example, under flipped label noise, noisy labels happen between similar classes (e.g., cat is more likely to be misclassified as dog than as flight), and it is likely that the same set of image features (e.g., object shape, tail length, etc) are used to classify an object as a cat or as a dog. In this case, $h(\cdot)$ can be viewed as the shared image features.
>
> [a] Sanjeev Arora, Simon Du, Wei Hu, Zhiyuan Li, and Ruosong Wang. Fine-grained analysis of optimization and generalization for overparameterized two-layer neural networks. In International Conference on Machine Learning, pp. 322–332, 2019a.
>
> [b] Sanjeev Arora, Simon S Du, Wei Hu, Zhiyuan Li, Russ R Salakhutdinov, and Ruosong Wang. On exact computation with an infinitely wide neural net. In Advances in Neural Information Processing Systems, pp. 8139–8148, 2019b.
>
> [c] Sanjeev Arora, Simon S. Du, Zhiyuan Li, Ruslan Salakhutdinov, Ruosong Wang, and Dingli Yu. Harnessing the power of infinitely wide deep nets on small-data tasks. In International Conference on Learning Representations, 2020.
>
> [d] Simon S Du, Kangcheng Hou, Russ R Salakhutdinov, Barnabas Poczos, Ruosong Wang, and Keyulu Xu. Graph neural tangent kernel: Fusing graph neural networks with graph kernels. In Advances in Neural Information Processing Systems, pp. 5724–5734, 2019b.
>
> >  "In the proof of Theorem 4, the authors state that ''we can ﬁnd a sub-structure (denoted as $\mathcal{N}_{sub}$ in the network $\mathcal{N}$ with sequential modules'' to learn the function $h$. However, definition 2 is used on clean data before. Is it an assumption on noisy data? If not, please justify it and add more discussions."
>
> We would like to draw the reviewer’s attention to assumption (a) in Theorem 4. Since $h$ is a function shared between the noise function $g(x) = g_r(h(x))$ and the target function $f(x) = f_r(h(x))$, the sub-structure $\mathcal{N}_{sub}$ can still learn $h$ effectively when trained with noisy labels.
>
> ---
>
> We hope our responses so far have cleared up the confusion for the reviewer to reevaluate and to discuss our contribution. We are happy to provide more context in the follow-up discussion. Please let us know if there is anything else we could clarify.

---

### Decision · Program_Chairs · 2021-09-28

**Decision:**

Accept (Poster)

**Comment:**

The paper studies the connection between the architecture of deep neural networks and their robustness to noisy labels, which hasn’t been studied yet. A take-away message is that a well-designed architecture can help learn good representations even the training sample have label noise. Theoretical and empirical analyses are provided. Although Reviewers RRS5 and NWJ7 concern that the theoretical analysis is limited, the paper has certain merits in empirical contributions. All the reviewers agree that the paper is interesting. One reviewer commented that the work can be potentially very useful in several areas, such as network architecture search, training with noisy data, and representation learning, to which the meta-reviewer also agrees. Since the paper brings a unique spark that may potentially enlighten other researchers and benefit the machine learning society, the meta-reviewer is happy to recommend an accept.

We ask the authors to carefully take the useful comments from reviewers in the final version, e.g., some writings that look overconfident should be revised. We also suggest the authors review an important topic in learning with noisy labels, i.e., modelling the label noise [r1], which has been exploited to correct loss [r2, r3] and should be useful in designing and validating the architecture of deep models by only exploiting the noisy data.

[r1] Yao et al. "Dual T: Reducing estimation error for transition matrix in label-noise learning."
In NeurIPS 2020.
[r2] Natarajan et al. "Learning with noisy labels." In NeurIPS 2013.
[r3] Liu et al. "Classification with noisy labels by importance reweighting." IEEE Transactions on pattern analysis and machine intelligence 38.3: 447-461, 2015.


**Consistency Experiment:**

NeurIPS has a long history of experimentation. In 2014, NeurIPS ran an experiment in which 10% of submissions were reviewed by two independent committees to quantify the randomness in the review process. This year, we repeated a variant of this experiment to see how the quality of the review process has changed over time.  This paper was part of the experiment and was therefore assigned to two committees (consisting of reviewers, an Area Chair, and a Senior Area Chair) that reached independent decisions.  If both committees made the same recommendation, this recommendation was followed. If a single committee recommended acceptance, the paper was accepted (with the exception of a few cases in which the other committee identified what we considered a fatal flaw, e.g., an error in a key result).

This copy’s committee reached the following decision: **Accept (Poster)**

The other committee assigned to the paper recommended **Reject**.  You can find the other set of reviews, along with any follow up discussion with the authors here:
https://openreview.net/forum?id=ViHTbcWJVv0